# A storm-relative climatology of compound hazards in Mediterranean cyclones

Raphaël Rousseau-Rizzi[1], Shira Raveh-Rubin[2], Jennifer Catto[3], Alice Portal[1], Yonatan Givon[2], and Olivia Martius[1]

[1]Institute of Geography, University of Bern, Hallerstrasse 12, CH-3012 Bern, Switzerland
[2]Department of Earth and Planetary Sciences, Weizmann Institute of Science, Rehovot 7610001, Israel
[3]College of Engineering, Mathematics and Physical Sciences, University of Exeter, Harrison Building, Streatham Campus, N Park Rd, Exeter, EX4 4QF, UK

**Correspondence:** Raphaël Rousseau-Rizzi (r.rousseaurizzi@gmail.com)

**Abstract.** Cyclones are responsible for much of the weather damage in the Mediterranean region and, while their association with individual weather hazards is well understood, their association with multivariate compound hazards remains to be quantified. Since hazard compounding is associated with enhanced risk, this study aims to establish a cyclone-relative climatology of three different multivariate hazards in Mediterranean cyclones, namely, the co-occurrences of rain and wind, rain and wave, and particulate matter and warm spells. The hazards are composited separately for nine cyclones classes associated with nine large-scale environments, using a recent cyclone PV-based classification. This cluster-based compositing of multivariate hazards outlines the role of the large-scale environment in the occurrence of impactful cyclones. The composites are computed relative to cyclones centers and at the time of maximum intensity, when the association with compound hazards is strongest for most of the nine cyclone classes, to illustrate the spatial footprint of the multivariate hazards associated with the cyclones. Finally, datasets of cold fronts, warm conveyor belts and dry intrusions are composited alongside the hazards to provide information on the contribution of smaller-scale features to the occurrence of multivariate hazards. We find that few different large-scale configurations are associated with each specific compound event type. Compound rain and wind events are mostly associated with frontal cyclones and cyclones induced by anticyclonic Rossby wave breaking. These events are most frequent in the winter half-year. Compound rain and wave events occur also primarily during winter, but are also associated with cyclonic Rossby wave breaking. Particulate matter and heat compound events are associated with heat lows, daughter cyclones and anticyclonic Rossby wave breaking in the warm season and over North-Africa. The probability of compounding associated with a cyclone class does not depend monotonically on the probabilities of the individual contributing hazards, but also on their temporal and spatial correspondence. Finally, we find warm conveyor belts and cold fronts to frequently co-occur with rain and wind, and rain and wave events. The association of compound hazards with warm conveyor belts and cold fronts is similar to previous results from the Atlantic basin but substantially modulated by the local topography and land-sea distribution. Particulate matter and warm spells are not strongly associated with these dynamical features. These results, which systematically associate various large-scale environments and dynamical features to different compound event types, have implications for forecasting and climate risk predictions.

## 1 Introduction

Mediterranean cyclones are extratropical cyclones that may involve tropical-like characteristics (Emanuel, 2005; Miglietta, 2019; Flaounas et al., 2022). They constitute a distinct group of storms which are typically smaller and weaker than their oceanic counterparts (Flaounas et al., 2015, 2022). Still, the Mediterranean basin is one of the most active cyclogenetic regions of the world (Ulbrich et al., 2009), and the Mediterranean cyclones are associated with most of the high wind and precipitation impact throughout the region (Flaounas et al., 2022). In addition to heavy precipitation (Pfahl and Wernli, 2012; Flaounas et al., 2018) and high winds (Nissen et al., 2014), Mediterranean cyclones are associated with a broad variety of hazards ranging from sea waves (Cavaleri et al., 2012; Patlakas et al., 2021) to the uplift and transport of particulate matter smaller than $10\ \mu m$ (PM10) (Dayan et al., 2008; Kalkstein et al., 2020; Nissenbaum et al., 2023), and impacting many aspects of society. Precipitation and waves for example, may impact society via flooding, coastal erosion and landslides.

Individually, such hazards may pose a risk to society, but recent research shows that the comparatively little studied compound events, may be responsible for an even larger fraction of the major climate and weather catastrophes impacting our societies (Zscheischler et al., 2018). Multivariate compound events, defined as events where the co-occurrence of hazards leads to enhanced risk of impacts by comparison to any individual hazard (Zscheischler et al., 2020), have been shown to occur in the Mediterranean. Among others, wind and precipitation compound events (Raveh-Rubin and Wernli, 2015; Martius et al.,

2016; Catto and Dowdy, 2021), sea waves and precipitation events (Bevacqua et al., 2019; Amores et al., 2020), and heat and PM10 events (Katsouyanni et al., 1993) have been observed in the region, and have been shown to impact infrastructure (wind, precipitation and waves) and public health (PM10 and heat). In this paper we will use the terminology multivariate hazards and multivariate compound events interchangeably to indicate the co-occurrence of hazards that have been shown by prior literature to lead to enhanced risk in the region. Contrarily to individual hazards, the relation between multivariate hazards

and Mediterranean cyclones has yet to be systematically characterized. Such a characterization is important, not only for the evaluation of climate risk (Bevacqua et al., 2019), but also to improve forecasting at shorter time scales (Saleh et al., 2017). We note that individual weather hazards may influence each other. For example, growing waves may lead to an increase in surface roughness and a decrease in near-surface wind speeds (Gentile et al., 2021).

Additionally, within individual cyclones, hazards tend to occur in relation with dynamical features such as cold fronts (CFs Dowdy and Catto, 2017), warm conveyor belts (WCBs, Flaounas et al., 2018) and dry intrusion (DIs, Raveh-Rubin, 2017). CFs are associated with 75% of the precipitation extremes in the midlatitudes (Catto and Pfahl, 2013) as well as with extreme winds (Dowdy and Catto, 2017; Catto and Dowdy, 2021), WCBs have been shown to be associated with extreme precipitation in the Mediterranean (Flaounas et al., 2018) and wind and wave near the British Isles (Gentile and Gray, 2023), and DIs are associ-

ated with enhanced surface wind gusts globally (Raveh-Rubin, 2017). Cyclone dynamical features and their relation to hazards in the Mediterranean may differ from those in other regions of the world because of the strong modulation of the cyclones by topography and by the land-sea distribution, and because of the distinct structures of the cyclones in the region (Flaounas et al., 2022). For example, case studies have shown lower cloudiness associated with WCBs in the Mediterranean, in cases where dry Saharan air was entrained in the cyclone (Ziv et al., 2010). Hence, quantifying the relation between Mediterranean

cyclone dynamical features and hazards provides valuable information on the smaller-scale processes involved in the occurrence of hazards. Similar to the relation between the hazards and the cyclones themselves, the role of dynamical features in the Mediterranean has been investigated for individual hazards but not yet systematically for compound events. Previous efforts to quantify the relation between multivariate compound events, extratropical cyclones and dynamical features elsewhere in the world (Owen, 2022) likely provide only partial information applicable to the Mediterranean basin. This is the case because

the sharp topography enclosing the small Mediterranean basin makes it a distinct and complex environment which cannot be directly compared to large ocean basins (Flaounas et al., 2022). Notably, an important cyclone characteristic which emerges from the Mediterranean basin configuration is that the genesis and intensity of Mediterranean cyclones is typically controlled by the upper-level potential vorticity (PV) structure (Flocas, 2000) even in the rare tropical-like cyclones, where low-level PV

generation is important as well (Flaounas et al., 2015). Even in less complex ocean basins, interactions accross scales mean that a dizzying variety of atmospheric configurations may give rise to high impact events. For that reason, a climatology of cyclone-related hazards in the Mediterranean must include information on the atmospheric configuration at multiple scales and levels, from upper-level synoptic scale flow, to surface dynamical features.

Hence, as a first step in characterizing the relation between compound hazards and Mediterranean cyclones, we aim to establish a storm-relative climatology of various compound events and the associated synoptic- and meso-scale dynamical features. Such a climatology

1. provides information on the spatial footprints of the compound hazards associated with the cyclones and on the seasonality of the link between cyclones and compound events;

2. clarifies the role of the large-scale dynamics in setting conditions conducive to the occurrence of particularly impactful cyclones; and

3. quantifies the relation between smaller-scale dynamical processes and the occurrence of compound events, to disentangle the contribution at various scales to the occurrence of compound events.

This Lagrangian climatology focuses on three types of multivariate compound events, namely the co-occurrences of strong winds and heavy precipitation, of heavy precipitation and high sea waves, and of high PM10 concentration and warm spells. Hereafter these will be referred to as "rain-wind", "rain-waves" and "PM10-heat" events. Studying these multivariate compound hazards requires evaluating the occurrence of five individual hazards within a single consistent framework, so this study also incidentally provides a novel unified comparison of multiple single hazards associated with Mediterranean cyclones. Hence the occurrence of these five individual hazards is also analyzed, albeit in less detail than the compound events. Finally, the occurrence of three different dynamical features is analyzed in relation to the hazards: cold fronts, warm conveyor belts and dry intrusions. For brevity, wind-wave events are not analyzed in this paper.

First, Sect.2 presents the methodology and data used in defining events and establishing the climatology, then Sect.3 presents summary results for the hazards and selected in depth analyses, and Sect.4 discusses mechanisms of compounding and compound event co-occurrence with dynamical features. A conclusion then synthesizes the study and introduces future areas of research.

## 2  Data and methods

The data required to establish this cyclone-relative climatology of compound hazards in Mediterranean cyclones is comprised of three different sets: 1) cyclone tracks to provide information on the location, timing and intensity of storms, 2) gridded datasets of individual hazards and 3) gridded datasets of dynamical features. Since all datasets, except PM10, are available for the period 1980-2019, all analyses cover that period, except for PM10-related analyses, which extend from 2004 to 2019.

The main methods required to process that data include 1) definitions of single-hazard and multivariate events, 2) compositing methods to characterize the occurrence of compound events and the co-occurence of events and dynamical features in the vicinity of storms, and 3) significance testing. The description of these datasets and methods follows.

## 2.1 Cyclone tracks and classification

To evaluate the relationship between compound hazards and Mediterranean cyclones, we use the storm tracks produced by the MedCyclones COST action, a coordinated initiative aiming to address unsolved questions regarding the dynamics, climatology and impacts of Mediterranean cyclones. This storm track dataset represents a best estimate of the cyclones' position and pressure intensity throughout their life cycle. The estimate is obtained by averaging multiple separate tracking methods (Flaounas et al., 2023), relying on variables such as pressure minima and gradients, low-level relative vorticity, and geopotential. The
tracks are computed using hourly ERA5 reanalysis data (Hersbach et al., 2020), the high spatial and temporal resolutions of which leads to accurate tracking of storm centers at the Mediterranean scale (Aragão and Porcù, 2022). Figure 1 shows the average track density per year of all the tracks of the MedCyclones dataset. The figure outlines hotspots of storm activity in the lee of the Alps (as identified by Buzzi and Tibaldi (1978)) as well as over the Sahara, in the lee of the Atlas, and south of the Anatolian mountains. In general, there is more tracked storm activity over bodies of water than over land, especially near the
Northern shore of the Mediterranean. The method used by Flaounas et al. (2023) to produce composite Mediterranean cyclone tracks requires the agreement of multiple (here chosen to be 5) different tracking methods. The fact that these methods tend to agree better over the sea than over land (Flaounas et al., 2023) partially explains the higher track density over bodies of water.

To clarify the large-scale context within which the cyclones and the associated compound events occur, we use the upper-level potential vorticity classification of Mediterranean cyclones introduced by Givon et al. (2024). In this classification, the cyclones are organized into nine clusters based on the surrounding PV field averaged over the 320-to-340 K dry isentropic levels at the time of minimum central pressure, using Self-Organizing Maps (SOMs, Kohonen, 1990). The similarity maps for all nine clusters, as well as the associated mean PV fields can be viewed in Figs. A1 and 3 of Givon et al. (2024). For conve-
nience, Fig. 3 of Givon et al. (2024) is reproduced in appendix as Fig. A1 of the present paper. In this PV-based classification, Cluster 1 cyclones are interpreted as reaching their minimum central pressure during stage A of lee-cyclogenesis (Mattocks and Bleck, 1986; Buzzi et al., 2020), with cluster 4 cyclones peaking at stage B. Mattocks and Bleck (1986) associated the rapid deepening of stage A to geostrophic adjustment processes in the lee of the Alps, and the consistent yet slower deepening in stage B to synoptic-scale baroclinicity, that can only contribute to cyclone intensification away from the topographic barrier.
As a result, while the cyclones in both clusters initiate in similar regions, cluster 4 (stage B) cyclones usually peak in a maritime environment further away from topography, present a strong frontal structure, and are usually deeper and more intense. Cluster 2 is considered to be a combination of Rossby wave anticyclonic and cyclonic wave breaking (AWB and CWB, Thorncroft et al., 1993), both of which have been shown to be involved in Mediterranean cyclones (Flocas, 2000). While cluster 5 represents "classic" AWB and is associated with weaker cyclones, Cluster 2 additionally features CWB near the tip of the PV

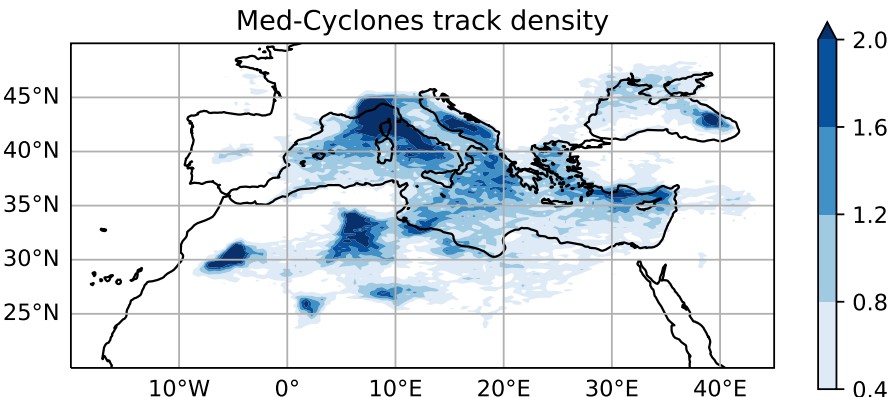

**Figure 1.** Density of cyclone tracks in the Mediterranean basin in units of number of cyclones per year per $1\times1$ degree box. Note that this represents only the average number of cyclone centers tracked at a given location per year, not the much higher average number of different cyclones which impact that location.

streamer. This explains the intensity of the cluster 2 cyclones because such a formation enables large PV values to penetrate furthest equatorward (Givon et al., 2024). Cluster 8 cyclones are somewhat weaker and may be associated with CWB. Cluster 7 cyclones are interpreted as daughter cyclones forming in the warm sector of parent cyclones located to their north (potentially cyclones from clusters 1 or 2). Finally, cluster 6 cyclones, which are very weakly baroclinic, are interpreted as heat lows, and clusters 3 and 9 are interpreted as cutoff lows. In the present study, only the storm clusters shown to be responsible for compound events will be discussed in detail.

## 2.2 Hazard datasets

The wind, precipitation, wave height and air temperature data used to define single- and compound hazards are obtained from ERA5 reanalysis (Hersbach et al., 2020), saved at 0.5 degrees resolution, but only 6h intervals for storage space economy. The wind variable is taken to be the magnitude of the 10-m wind components, which includes eastward and northward winds ("u10" and "v10" respectively in ERA5). We note that these represents grid-cell average winds which have smaller variability than observations. The ERA5 total precipitation variable used ("TP"), represents the 1-hour accumulated grid-cell average precipitation. This variable does not directly assimilate precipitation observations, but instead represents a short-term forecast initialized by the reanalysis (Hennermann and Berrisford, 2020), which has been shown to compare favorably to extreme precipitation in the mid-latitudes (Rivoire et al., 2021). The wave variable used here is the "significant wave height" ("swh"), which represents the height between the wave crests and troughs and is defined as the square root of the integral in space and frequency of the surface wave spectrum, times four. This variable can serve to assess the damage and flooding potential

of the waves. The reanalysis swh also compares favorably to observations, albeit ERA5 may underestimate extreme wave heights near the coasts (Fanti et al., 2023), which are of interest here. Next, the temperature variable considered to assess the compounded impacts of dust and heat on human health is the temperature at two meters above ground ("t2m"). Finally, we use near-surface particulate matter at diameters lower than 10 $\mu m$ ("PM10"), provided by the Copernicus Atmosphere Monitoring Service (CAMS) reanalysis (Inness et al., 2019) at a resolution of 0.75 degrees. The CAMS PM10 variable compares very well with surface observations of particulate matter over Europe (Rémy et al., 2019) and North Africa (Fluck and Raveh-Rubin, 2023). We note that, even though PM10 combines multiple aerosol types at diameters less than 10 $\mu m$, the majority of the concentration around the Mediterranean basin is due to dust particles, the main source of which is the Sahara (Rémy et al., 2019). Hence, here we will consider that the origin of high PM10 events is usually the emission and transport of Saharan dust.

## 2.3 Dynamical features datasets

The dynamical features datasets used here were introduced in previous studies, and kindly made available to us. These include a dataset of objectively identified front lines (Sansom and Catto, 2024), a dataset of DI trajectory density (Raveh-Rubin, 2017) and a dataset of WCB trajectory density (Heitmann et al., 2023), computed using ERA5. The front and DI datasets were initially introduced based on ERA-Interim data; we are using here an updated version, based on ERA5. The DI identification is based on a systematic airmass trajectory computation, where DI trajectories are defined by their descending of at least 400 hPa within 48 h. A DI is then considered present at a given grid point and time if any DI trajectory is present there at a pressure level of 700hPa or higher (i.e., in the lower troposphere). Thus, we consider so-called "DI outflows" as referred to in Catto and Raveh-Rubin (2019) and adapted here to ERA5. Similarly, the WCB identification is based on trajectories that ascend 600 hPa or more within 48 h. The density of WCB trajectories is then computed for three different pressure intervals (Heitmann et al., 2023): an inflow interval (between the surface and 800 hPa), an ascent interval (between 800 and 400 hPa), and an outflow interval (pressures lower than 400 hPa). While the methods used to identify DIs and WCBs are similar, the front identification method (based on Hewson, 1998) is very distinct. The method first identifies contours where the gradient of the thermal front parameter (Renard and Clarke, 1965) based on wet bulb potential temperature is zero, and then masks the regions according to thresholds of the gradient of the wet bulb potential temperature across the front and in the adjacent baroclinic zone (Sansom and Catto, 2024). As in the method of Catto and Pfahl (2013), cold fronts are then identified based on the front speed and direction.

## 2.4 Event definition

In this study, the occurrence of a multivariate compound event is defined as the co-occurrence of two single-hazard events, as was done in previous literature such as by Owen et al. (2021). Single-hazard events are defined as the exceedance of a threshold chosen to represent a given level of risk. Risk depends not only upon the magnitude of the hazard but also upon the local vulnerability to that hazard, which may have large spatial variability. When that is the case, a given hazard magnitude may be benign in one region and catastrophic in another, and local percentile thresholds are most appropriate to capture the local risk. Otherwise, a fixed threshold may be used to define the occurrence of events everywhere.

The vulnerability to the precipitation hazard is highly variable spatially and, following previous literature (e.g., Catto and Pfahl, 2013), we choose the $99^{th}$ percentile as a threshold for the occurrence of an event. In addition, since our analysis domain includes very arid regions like the Sahara, we seek to avoid considering trace precipitation as rain events. To do so, we add another criterion to the event threshold, which is that the 1-hour integrated precipitation must also exceed 1 mm to be considered an event. In other words $T_p = \max(99^{th}, 1 \text{ mm/h})$, where $T_p$ is the precipitation event threshold. Similarly, the vulnerability to wind damage is variable in space, and in keeping with previous literature, we select the $98^{th}$ percentile of wind magnitude (Klawa and Ulbrich, 2003), along with a fixed minimum threshold of 10 m s$^{-1}$ to yield a wind threshold $T_w = \max(98^{th}, 10 \text{ m s}^{-1})$. Figure 2 shows maps of the precipitation and wind thresholds used in this study. The top panel shows that wind thresholds are highest over bodies of water and coastlines, except in particularly windy locations such as Chad (at the Bodele depression) or Iceland. The bottom panel shows that precipitation thresholds are typically highest downwind of large bodies of water, in mountainous regions and, within the Mediterranean Basin, wherever Mediterranean cyclone track density is high (see Fig. 1). At the same time, they are very low in subtropical subsidence regions such as the Sahara and the Arabian Peninsula. Note that the domain presented in Fig. 2 is larger than that within which Mediterranean cyclones are tracked (shown in Fig. 1) so that compositing is possible around cyclones occurring on the edge of the Mediterranean basin.

The threshold for waves is taken here to be fixed at 4 m because that is a representative value for the $99^{th}$ percentile of significant wave height (e.g., Barbariol et al., 2021, Fig.5) and close to the significant wave height observed in damaging storms (e.g., Amores et al., 2020). This is reasonable since there does not seem to be an agreed-upon definition of space-varying wave event threshold, and percentile maps are not as variable for waves as they are for precipitation or winds. In the case of PM10, high concentrations may pose health risks anywhere and a fixed threshold is chosen. We select a 50 $\mu$g m$^{-3}$, 24-h averaged threshold, equivalent to the recommendation of the European union for 24h exposure to PM10 (EU and Parliament, 2008) and 10% higher than the World Health Organization recommendation (WHO and for Environment, 2021). Finally, for warm spells, we chose a fixed 30 °C threshold to be consistent with previous studies of increased mortality in the event of co-occurring dust and heat extremes (Katsouyanni et al., 1993). We note that since empirical quantiles are not defined for multivariate distributions, defining compound events as the co-occurrence of single-hazard events circumvents having to fit a copula to the multivariate distribution of hazards, and hence offers a bit more flexibility in defining such events.

### 2.4.1 Dynamical features occurrence

The criterion to define whether warm conveyor belts or dry intrusions are present at a given gridpoint is taken to be the presence of a non-zero WCB or DI trajectory density at that gridpoint. Here, WCB trajectories are assumed to be important for surface hazards only when they occur in the "inflow" or "ascent" regions of the conveyor belt (between the surface and a pressure of 400 hPa). The inflow region is expected to be associated with enhanced surface winds, and the ascent region, with enhanced precipitation. In this study, no distinction is made between the two regions considered. WCB and DI trajectory density data is

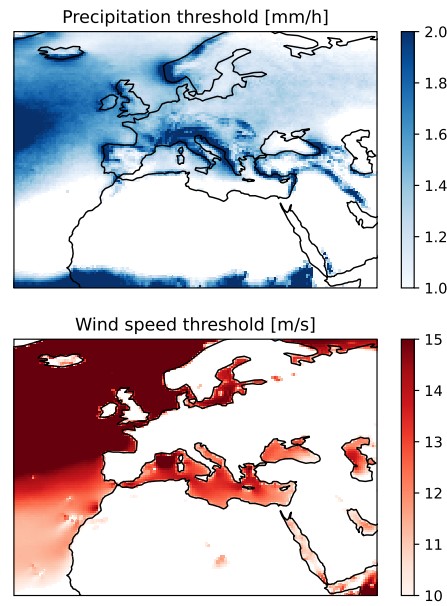

**Figure 2.** Maps of precipitation and wind thresholds

2-dimensional (2D), so it can be directly compared to 2D hazard data. On the other hand, in the dataset we are using, fronts are defined as 1D features (Catto and Pfahl, 2013). To assess the role that fronts may have in the occurrence of hazards, a "zone of influence" must be defined and the 1D features must be transformed into 2D maps. Following previous literature (e.g., Catto and Pfahl, 2013) any location within 2.5 degrees of a point where a front is located is deemed to be potentially influenced by that front.

## 2.5  Compositing

The cyclone-relative composites are obtained by first evaluating the occurrence of single-hazard events for all variables, in a 40×40 degrees box centered on each cyclone, only at the time of minimum central pressure. This large box was chosen because the scale of the significant hazard fields is not a priori known for all hazards, and to produce validation composites of the large-scale environment (not shown). Since the time resolution of the storm tracks is sometimes higher than that of the hazard data, we select the minimum pressure over the times when hazard data is available. Selecting a single snapshot per storm removes any ambiguity in separating events or in evaluating the contribution of each storm to the final composite (Owen, 2022). Selecting specifically for the time of minimum pressure, however, makes the hypothesis that, at that time, compound hazard probability is at a maximum. This hypothesis will be verified in Sect. 3. The hazard data is linearly interpolated onto

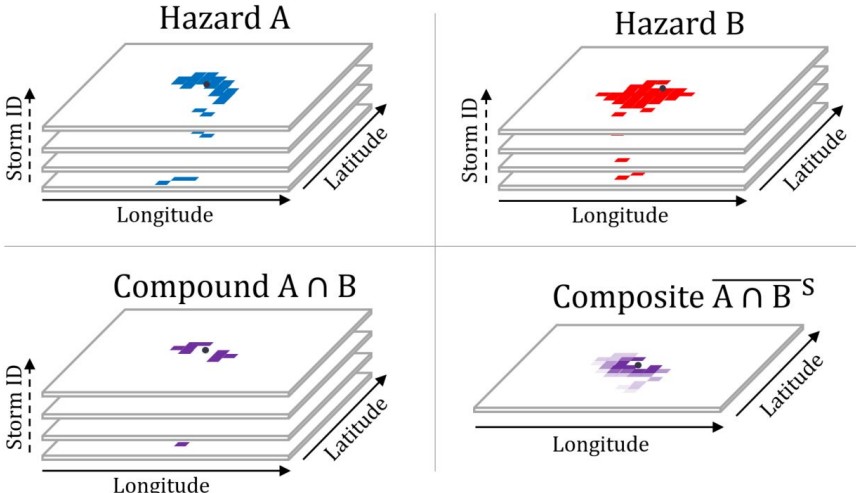

**Figure 3.** Boolean matrices of event occurrence for individual hazards (Hazards A and B), with dimensions Storm ID, and storm-relative horizontal coordinates. Colors (blue and red) indicate event occurrence (1) and white space indicate non-occurrence (0). Boolean matrix of compound events (Compound $A \cap B$) obtained by taking the intersection of Hazard A and Hazard B (purple). Matrix of compound event probabilities (Composite $\overline{A \cap B}^S$) obtained by averaging the compound event matrix across the Storm ID dimension. Darker shades of purple indicate higher probabilities (maximum 1) and lighter shades indicate lower probabilities (minimum 0).

the storm box, at a 0.25 degrees resolution, as is the hazard threshold if it is variable in space (precipitation and winds).
Event occurrence is then computed, yielding one 2D boolean matrix per storm and per hazard, with storm-relative latitude and longitude, indicating where an event is detected (value of 1) and where no event is detected (value of 0). Each individual storm matrix is attributed an index (Storm ID). These matrices are produced for all cyclones and then stacked into one 3D boolean matrix per hazard with dimensions Storm ID, storm-relative longitude and storm-relative latitude. This is illustrated on the top row of Fig. 3, where colored cells indicate hazard threshold exceedance (value of 1) and white cells indicate no threshold
exceedance (value of 0). The composites of individual hazard frequency are obtained by averaging these matrices along the Storm ID dimension.

### 2.5.1 Co-occurrence analysis

To evaluate the occurrence of multivariate compound hazards, one simply needs to compute the co-occurrence of individual hazards by taking the intersection of individual hazard boolean matrices. In Fig. 3, the bottom left panel illustrates the result of
245 computing the compounding of Hazard A and Hazard B (top row). The main thing to notice is that the area of compound events is smaller than that of either individual hazard and that its shape is determined by the overlap of the individual hazard footprints. The final step to obtain storm-relative composites of compound events is to take the average of the boolean compound matrix along the Storm ID dimension (Fig. 3, bottom right). This yields storm-relative maps of compound event occurrence probabilities, with values between 0 (never occurs in the storm sample) and 1 (always occurs in the storm sample).

 ## 2.5.2 Masked compositing

In the case where we composite hazards which are not defined everywhere, such as waves which only happen in the sea, we need to proceed a bit differently. Storms are only included in the composite if their centers are over the Mediterranean Sea at the time of minimum pressure, and all land points as well as the Atlantic Ocean are masked out and are discounted in the averaging of the 3D occurrence matrices to obtain composites. This means that a spatially varying number of different storms is used in computing the final storm composite map. The number of data points available for compositing tends to decrease away from the storm center, which mean that the data there is less reliable. This is accounted for in the significance testing; locations where there are too few points are not found to be significant.

## 2.6 Significance testing

The compositing step yields a storm-relative map of compound event probabilities. To assess the statistical significance of each point constituting the map, we aim to test the null hypothesis that the probabilities of compounding associated with storm events is not different from the probabilities of compound events at all times at that point. To that end, we produce Monte-Carlo samples of hazard events at the same locations as those of the storm samples, but at random times. Those times are selected for random years but within 15 days of the days of year where the storm events occur, to yield the same seasonal distribution as that of the storm sample (e.g., Welker et al., 2014). We produce 2000 such Monte-Carlo samples for each hazard, which are used to compute 2000 maps of compound hazard probabilities. Empirical p-values are then computed at each point of the storm-relative maps of compound hazard probabilities as one minus the quantile of that point within the corresponding Monte-Carlo samples. If the probability at the storm point is the largest, it is attributed a p-value of 0. When evaluating a large number of significance tests, it becomes problematic to reject the null hypothesis based upon a fixed threshold for the p-value (Wilks, 2016). Hence, following Wilks (2016), we control for the false discovery rate by sorting all $N$ empirical p-values in ascending order, from $p_1$ to $p_N$ and by computing an effective p-value significance threshold $p_{fdr}$ given by

$$p_{fdr} = \max_i \left( p_i \quad \middle| \quad p_i \leq \alpha \frac{i}{N} \right) \tag{1}$$

where $\alpha = 0.05$ is the false discovery rate control level (Wilks, 2016)). In practice, in our analyses, $p_{fdr}$ is around 0.005, which is considerably more constraining than a typical 0.05 threshold.

## 3 Results

For brevity, we do not show the storm-relative footprints of five single hazards and three types of multivariate compounding for all nine storm clusters. We first present compact summaries of the risk associated with each hazard and each compounding type in each cluster. These summaries are then used to determine which storm clusters are worthy of note in the context of the different types of multivariate compounding analyzed here. Once the most important cyclone clusters are identified, their compounding structure as well as their large-scale context is discussed. To summarize the risk associated with a (single or

compound) hazard within a given cluster, we average the hazard probabilities over a 20×20 degrees box centered on the storm composite for that cluster. Probabilities that are not statistically significant are set to zero in this averaging process. In that analysis, storms that either have points with higher hazard probabilities or a larger hazard footprint are found to be associated with higher risk. Setting the probabilities of non-significant points to zero ensures that clusters for which the probability of a hazard has no field significance will be found to have zero risk associated with that hazard. If we consider Boolean matrices as binary matrices, the summary metrics for single and compound hazards are computed as

$$P_A = \overline{A}^{SH}, \quad P_C = \overline{\min(A,B)}^{SH}, \tag{2}$$

Where A and B are two different hazards, the overline denotes an average, and $S$ and $H$ are respectively the "Storm ID" and the horizontal (latitude-longitude) dimensions of a 20×20 degree box around the cluster center. $P_A$ and $P_C$ are respectively the probabilities of hazard A and compound hazard A-B, which may be interpreted as the fraction of all cases where a compound event will occur at any point within a 20×20 degrees box centered on a Mediterranean cyclone. Note that the probabilities of different hazards or clusters are more meaningful relative to one-another than considered independently because the size of the averaging box influences their absolute value. Doubling the area of the averaging box would lead to a halving of all average probabilities, as long as all statistically significant points are encompassed within both boxes. We also note that the choice of cyclone-relative latitude and longitude coordinates means that the size of the area changes between events depending on the latitudinal position of the cyclones. This approach is chosen to ensure comparability with other studies (e.g., Givon et al., 2024).

### 3.1 Individual and compound hazards overview

First, Fig. 4 shows the averaged probability of occurrence for the rain hazard, the wind hazard and the compounding of both winds and rain. Each bar of the plot represents a different cluster, and the clusters are sorted by average compound risk, in decreasing order. The average compound event probability exhibits large differences between clusters, with three clusters (2, 4 and 1) being responsible for the majority of the rain and wind compound risk. These clusters are associated with higher compound risk to infrastructure and will be investigated in more detail. Cluster 6 is not associated with any statistically significant occurrence of compound events despite being associated with small but statistically significant rain events and wind events. This leads us to the observation that rain-wind compound probabilities are systematically multiple times smaller than either rain probabilities or wind probabilities alone. Further, while the probability of compounding scales approximately with the probabilities of individual hazards, there are cases where lower probabilities of individual hazards are associated with higher probabilities of compounding. Notably, cluster 2 which exhibits the highest compounding probabilities, has lower individual probabilities of both precipitation and wind events than cluster 4. We will return to this result in Sect.4.

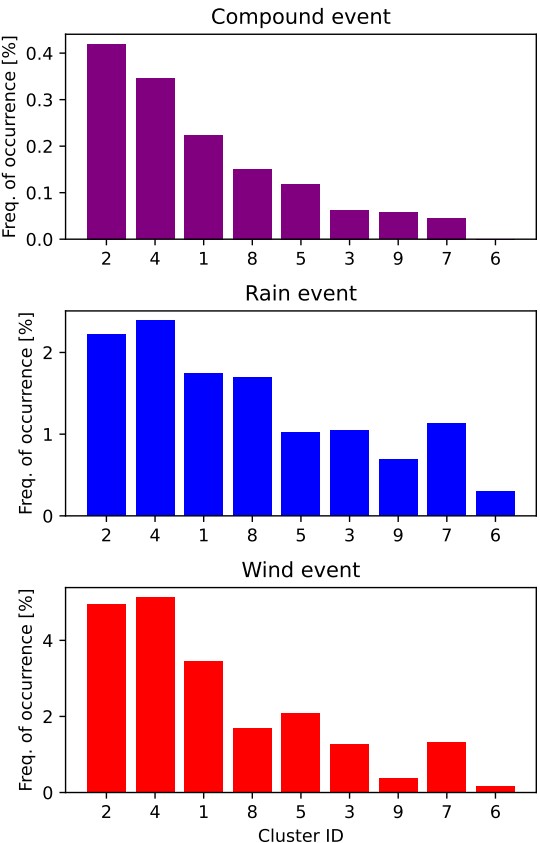

**Figure 4.** Risk summary for all clusters for rain and wind compounding (top panel, purple bars), for rain alone (middle panel, blue bars) and for wind alone (bottom panel, red bars). The clusters are plotted in decreasing order of compound risk, for all panels.

Next, Fig. 5 shows the averaged probability of rain hazard, wave hazard and the compounding of both rain and waves. Note that the rain probabilities are computed only over the sea here and, in order to be more directly comparable, are masked in the same way as the wave events. The masked composites for clusters 2, 4 and 8 include respectively 65%, 52% and 39% percents of the total number of cyclones in the clusters, distributed between 1980 and 2019. As a result, the rain probabilities shown here are slightly different than the ones shown in Fig. 4. For example cluster 6 is not associated with any statistically significant precipitation over the sea because storms in that cluster occur almost exclusively over land (Givon et al., 2024). Here, only three clusters (2, 4 and 8) are associated with statistically significant compound flooding risk. Note that cluster 1 compound event probability is not significant despite the associated wave event probability being higher than that of cluster 8.

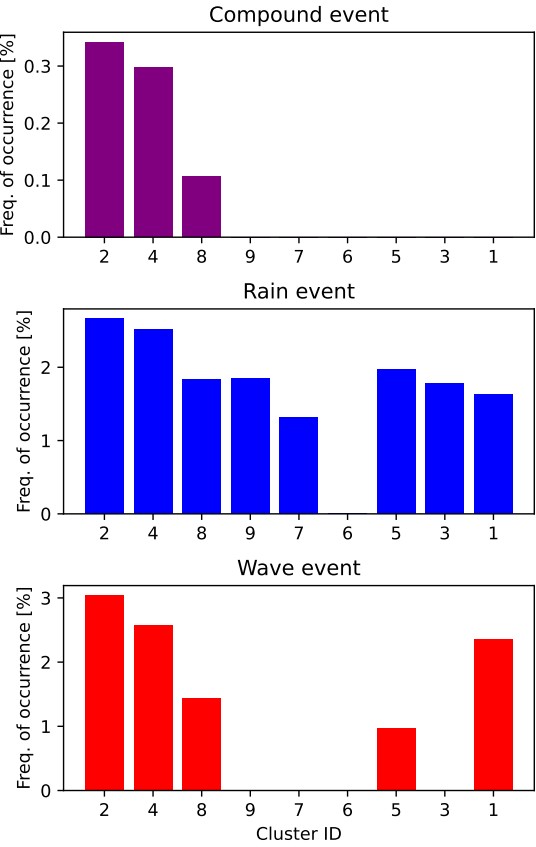

**Figure 5.** Risk summary for all clusters for rain and waves compounding (top panel, purple bars), for rain alone (middle panel, blue bars) and for waves alone (bottom panel, red bars). The clusters are plotted in decreasing order of compound risk, for all panels.

Finally, Fig. 6 shows the averaged probabilities of PM10 hazard, heat hazard and the compounding of both PM10 and heat. Once again, most of the compound events are concentrated within a few clusters which pose a higher risk to public health (6, 7 and 5), with some clusters exhibiting no significant compositing (2, 4 and 1). This time however, by contrast to the rain-wind and rain-waves compound types, there is no clear scaling of the compound event probability with the individual hazard probabilities. Note the case of cluster 4 which is associated with the highest PM10 probability but no statistically significant heat or compound event probability. Interestingly, there is little overlap between the clusters that are important for PM10-heat

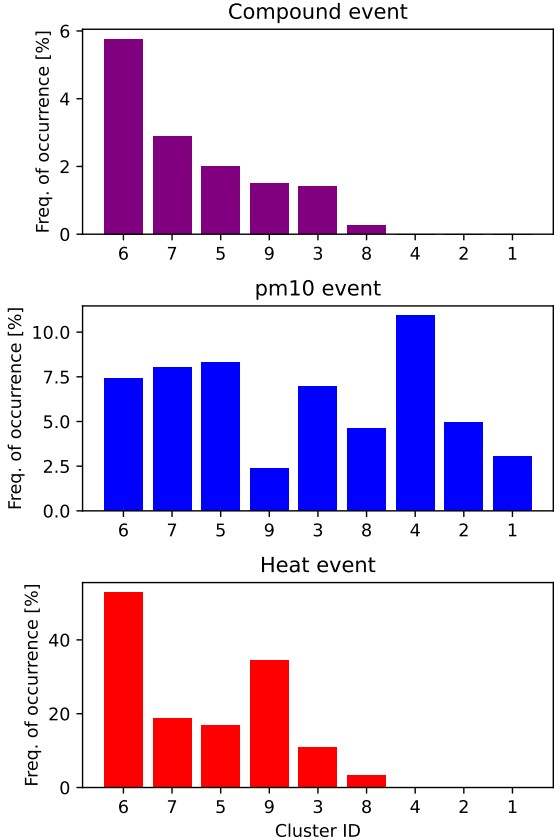

**Figure 6.** Risk summary for all clusters for PM10 and heat compounding (top panel, purple bars), for PM10 alone (middle panel, blue bars) and for heat alone (bottom panel, red bars). The clusters are plotted in decreasing order of compound risk, for all panels.

events, and the clusters responsible for both rain-wind events and rain-wave events. In fact, none of the clusters most important for wind and rain compositing (2, 4 and 1) are associated with warm spell field significance.

Based on the summary of compound event probabilities (Figs.4-6), we select for further analysis the three clusters which are most important for each type of compounding. Weighting by the relative number of storms tracked in each cluster (Givon et al., 2024), the clusters 2, 4 and 1 collectively account for 79% of rain-wind events, cluster 2, 4 and 8 account for all rain-wave events, and clusters 6, 7 and 5 account for 80% of PM10-heat events. These high percentages warrant focusing further analyses on the three clusters selected as most important for each compounding type.

## 3.2 Geographical, seasonal and dynamical context of clusters of interest

Figure 7 shows the relative density and seasonal cycle of cyclones at the time of minimum central pressure (used in compositing), which provides insight on the geographical and seasonal context within which cyclones associated with compound events occur. The relative density is computed by dividing the number of cyclones occurring over a given area by the total number of cyclones in the clusters considered. As can be expected, the density plots capture similar high-density areas as Fig. 1, which represented the density of tracks at all times. First, the density of events associated with rain-wind compounding is highest around Corsica and northern Italy, and is generally high along the north shore of the Mediterranean. The absence of rain-wind compound storms in the south is consistent with the seasonal cycle of these cyclones, which peaks markedly in winter, and reaches a minimum in mid-to-late summer. Since there is a single different cluster, the geographical and seasonal distribution of rain-wave storms is very similar to that of rain-wind storms. The main spatial density differences are that rain-wave storms are a bit less dense around northern Italy, and a bit more around Cyprus, as well as occurring occasionally further south. The seasonal cycle of rain-wave storms plateaus in winter and early spring before decreasing to a minimum in summer and re-increasing through fall. Finally, PM10-heat storms occur much more to the south than other clusters, especially over Morocco and Algeria, but still occur occasionally throughout the Mediterranean basin. Correspondingly, these cyclones occur least in winter, increase in early spring, plateau through summer, and decrease during the fall. The differences in seasonality and spatial distribution of the clusters associated with rain-wind and rain wave events on the one hand, and PM10-heat events on the other hand warrants the partition of Mediterranean cyclones into clusters, which can be studied separately for more accurate insight into specific compound hazards.

Recall that clusters 1 and 4 stage A and B of lee cyclogenesis respectively, which is consistent with their seasonal cycle peaking in winter and their geographical distribution. Clusters 5 and 8 represent AWB and CWB respectively, while cluster 2 represents a combination of both AWB and CWB, resulting in stronger cyclones. The small-scale and fast-moving daughter cyclones of cluster 7 often form in the lee of the Atlas during transition seasons, and are associated with dust storms (Bou Karam et al., 2010). At the same time the parent cyclones located to their north may belong to clusters 1 or 2. This is particularly interesting because it outlines that, in transition seasons, a given large-scale configuration may give rise simultaneously to rain-wind and PM10-heat events in different locations. Finally, cluster 6 heat lows occur mainly in the Sahara in summer. To sum up, rain-wind events are predominantly associated with anticyclonic wave breaking and type A and B frontal cyclones, rain-wave events are additionally associated with cyclonic wave breaking, and PM10-heat events are associated with heat lows, daughter cyclones and anticyclonic wave breaking in the subtropics. Note that there are strong seasonal and geographic variations in the PV structure associated with cyclones in the Mediterranean (Givon et al., 2024). For that reason, while they outline the role of dynamics more, the results obtained using a PV-based classification are likely similar to the results one would obtain using geographical and seasonal classifications, and must be interpreted in that context. For example, despite similar upper level

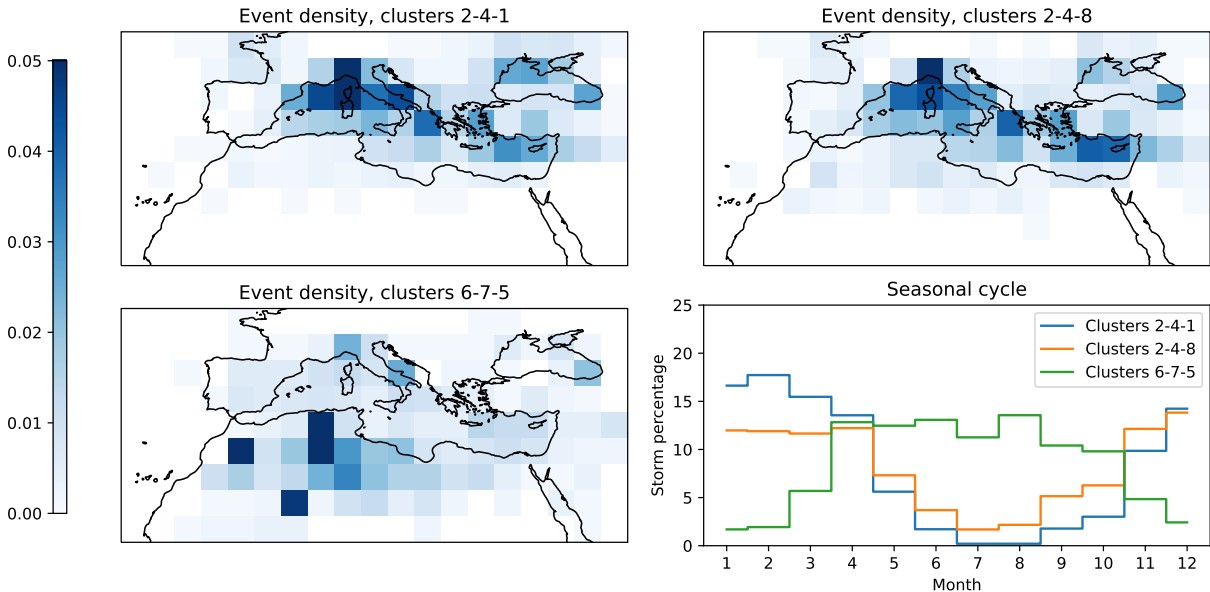

**Figure 7.** Relative spatial density of cyclones at the time of minimum central pressure for the clusters associated with rain-wind events (top left, clusters 2-4-1), with rain-wave events (top right, clusters 2-4-8) and with PM10-heat events (bottom left, clusters 6-7-5). Seasonal cycles of cyclones associated with the three compounding types (bottom right).

configurations, the very different regions within which cluster 2 and 5 storms occur largely influence the hazards with which they are associated (PM10 and heat for cluster 5, and wind, waves and precipitation for cluster 2).

### 3.3 Time evolution of clusters of interest

Having identified clusters of interest and determined their seasonal and spatial distributions, we next turn to the time evolution of compound hazards for these clusters, around the time of minimum central pressure. In addition to providing some insight on the compound hazards, this allows us to verify the hypothesis that the time of minimum central pressure is representative of the time of maximum compound hazard probability. Because the field statistical significance analysis computation is costly, the analysis is a little bit simplified here. Hence, Fig. 8 shows the time evolution of the fraction of storms where compound

events happen at any point within a 1000 km radius, and of the average percentage of the area occupied by compound events within this radius. Values are computed every 6 hours, from 24 hours before to 24 hours after the time of minimum central pressure. Both the fraction of storms and the average area percentage tend to vary similarly. First, we note that the average area percentages of Rain-Wind and Rain-Wave events do tend to increase, peak around the time of minimum central pressure, and then decrease markedly. The area peak is 6 hours before minimum pressure in Rain-Wind cluster 1 and 6 hours after in Rain-

Wave clusters 2 and 8, but we think that the differences with the time of minimum pressure are sufficiently small to warrant

using the time of minimum pressure to represent the maximum compound event likelihood. The fraction of storms where compound events occur does tend to systematically peak shortly after the time of minimum pressure in Rain-Wave events, but that variable is less representative of the risk posed by these storms than the average area percentage. The PM10-heat clusters tell a different story. While the time of minimum central pressure represents a maximum of average compound event area percentage, that maximum is local rather than global. Indeed, a very clear 24h cycle is visible, which indicates that the time of minimum pressure itself is, in these clusters, aligned with the diurnal cycle. This suggests that the diurnal cycle controls strongly both the temperature of the warm spells and the storm central pressure. By extension, the winds the storm induces, which are responsible for dust (PM10) lofting are also controlled by the diurnal cycle. This interesting result is consistent with dry cyclones occurring over the desert, where the diurnal cycle is very pronounced. Despite there being multiple maxima in the PM10-heat clusters, no time presents a much higher event probability than the time of minimum central pressure, which reaffirms this choice as a reasonable and representative compositing time step.

## 3.4 Hazard footprints

Having discussed the context within which storms associated with compound event occur, we now examine the hazard footprint of the cyclones within clusters of interest. Starting with rain-wind events, Fig. 9 shows the storm-relative composite probability of occurrence for rain, wind and rain-wind events. Color shading shows the percentage of storms where a hazard event occurs at a given location and in a given cluster, at the time of minimum pressure. The black contour encloses the statistically significant hazard area. As reported in previous literature (Owen, 2022), the distribution of individual events around the storm centers differs quite strongly between wind and rain events. Rain events are mostly concentrated to the north of the storm centers and the highest event probability is located to the northwest of the center. The wind event footprint, on the other hand, is much broader than that of rain, and mostly concentrated south of the center. Wind events occur most frequently south-south-west of the storm centers, and almost never immediately near the storm center. This sparsity of rain events around the storm centers in our composites shows that the location of the tracked center in the dataset used here (Flaounas et al., 2023) coincides well with the quiescent storm centers in ERA5 data. In clusters 1 and 4, the location of the wind maximum is more directly to the south of the storm center than reported for Atlantic cyclones (e.g., Field and Wood, 2007; Owen, 2022), which resembles most the wind signature of cluster 2. The wind signature of subsets of Atlantic cyclones occurring over central Europe may be more similar to that of clusters 1 and 4 (Eisenstein et al., 2023). The probability of occurrence of events as defined here (see Sect.2) peaks at around 24% for rain and 30% for winds. The footprints also vary in both shape and magnitude across clusters. Cluster 4, has an intense and broad rain footprint while cluster 2 has a smaller but equally intense footprint which is more concentrated to the northwest. Cluster 1 has the smallest and weakest rain footprint, but the statistically significant rain risk area extends further south than that of the other clusters. Wind events are most frequent and most concentrated to the south in cluster 4, while in cluster 2, and to a lesser extent cluster 1, the wind footprint wraps around the storm center to the west and up to the northwest. As a consequence of the very distinct areas occupied by the wind and rain event, compound events tend to occur less frequently than either individual events (peaking around a probability of 5%), and to be concentrated where both footprints overlap the most. In general, compound risk peaks to the west-north-west of the storm center, with a crown of (lower) risk

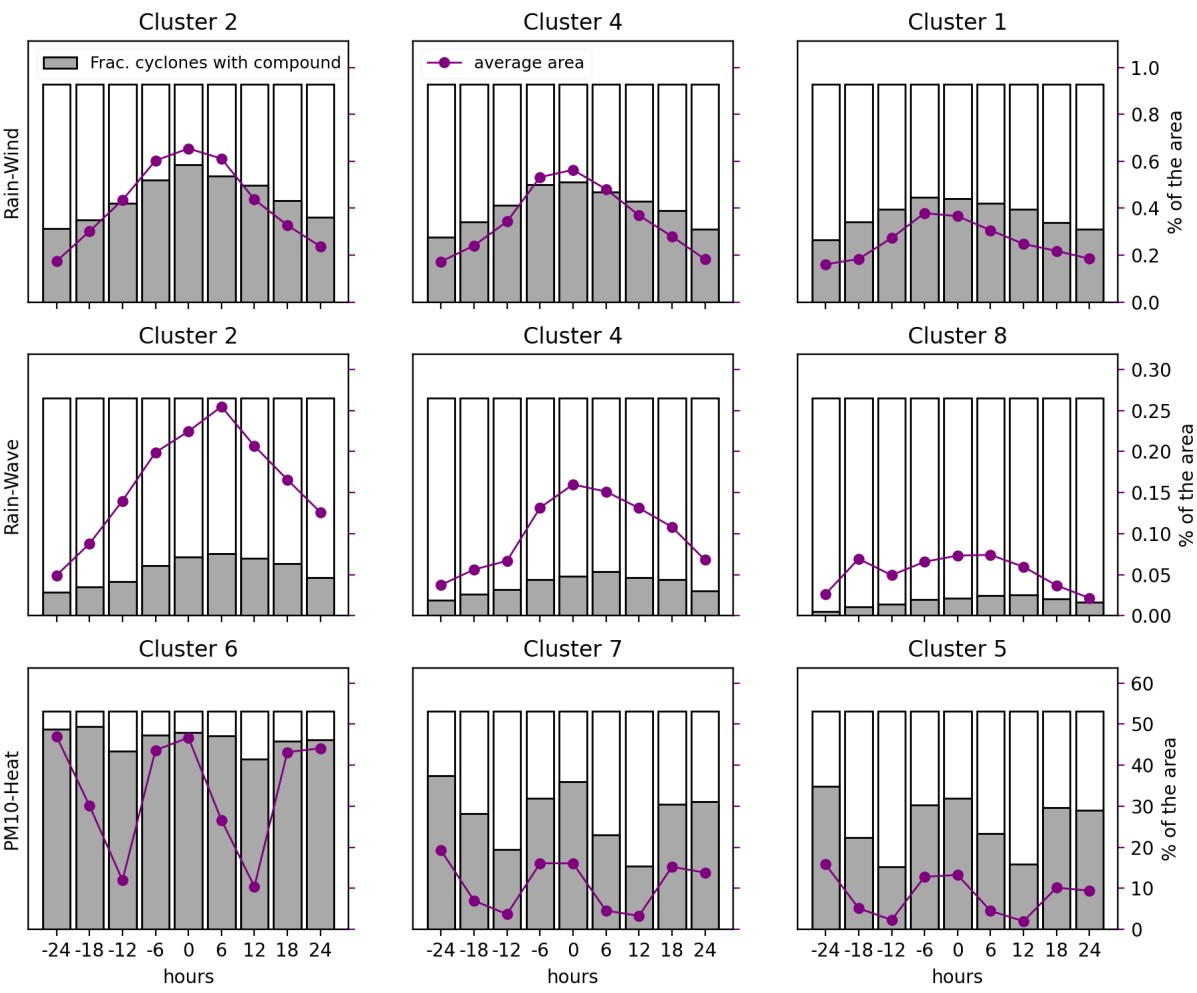

**Figure 8.** Time evolution of composites over a 1000 km radius disk around storm centers for Rain-Wind (top row), Rain-Wave (middle row) and PM10-Heat (bottom row) clusters of interest: Average percentage of the area that is occupied by compound hazards (purple dots and line) and fraction of tracked cyclones with compound hazards (gray bars). The fraction of cyclones with compound hazards is represented by the fraction of the total bar height (gray + white) that is occupied by the gray bar.

encircling the storm center. Despite having somewhat smaller and weaker wind and rain footprints than cluster 4, cluster 2 is associated with the highest rain-wind compound risk as noted in section 3.1.

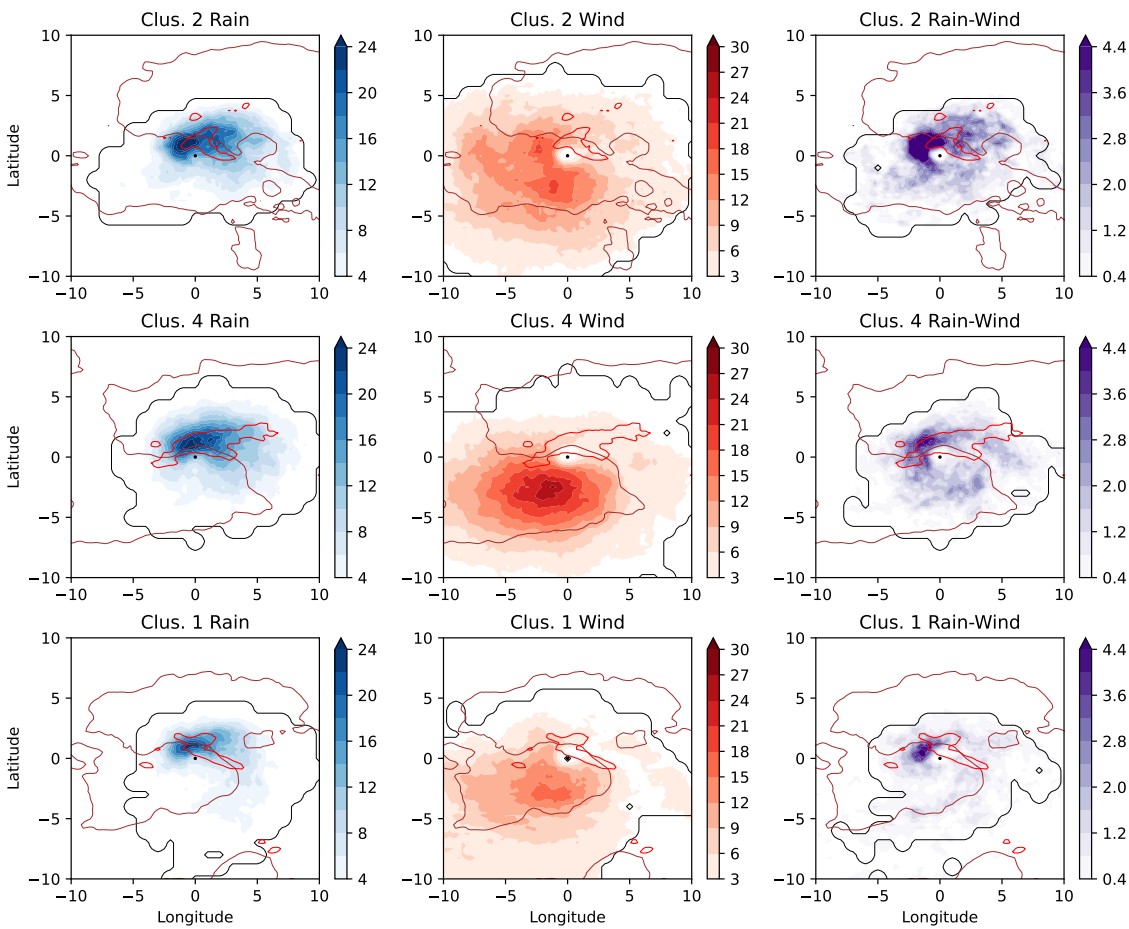

**Figure 9.** Storm-relative composite probabilities (shading) of rain events (blue, left column), wind events (red, center column) and compound rain-wind events (purple, right column) and for cluster 2 (top row), cluster 4 (middle row) and cluster 1 (bottom row). Probabilities are in %. The black contour identifies the statistically significant area for each hazard (see Eq.1). The brown and red contours respectively identify the 300 m isohypse and the 100-m-per-degree meridional gradient of the storm-relative elevation composite.

Next, Fig. 10 shows the storm-relative composite probability of occurrence for rain, wave and rain-wave events. These composites are computed over the Mediterranean sea only. Both the land and the Atlantic Ocean are masked out from the averaging, and only storms that are over the sea at the time of minimum pressure are averaged in the composite. Masking out land and the ocean means that a small number of data points are used in the composites far from the storm center. There, the composite becomes very noisy, and hence, for ease of interpretation, locations where the hazard probabilities are not significant are left

blank. Similarly to the rain and wind events, rain and wave events footprints vary widely between hazards and across clusters. Rain events over the sea tend to occur mostly north of the storm center, as is the case when rain over land is also considered (see Fig. 4). Wave events occur most frequently to the west of storm centers. In clusters 2 and 4, the wave event footprints tend to extend to the west, but also to the northeast of the storm center. As a result, in these clusters, waves and rain events also co-occur most to the north of the event. Wave event probability peaks around 16% while rain-wave event probability peaks around 6%. Interestingly, the composite of rain events occurring only over sea does not exhibit much of a magnitude difference with the composite of all rain events. This can also be seen by comparing the "Rain event" graphs in Fig. 4 and Fig. 5.

Finally, Fig. 11 shows storm-relative composites of PM10, heat and PM10-heat events. The PM10 events are fairly well distributed around cluster 6 cyclone centers, but peak a bit to the northeast. In clusters 7 and 5, however, the PM10 event probability footprint is located clearly to the south of the storm center, and stretches along a southwest-to-northeast axis reminiscent of a trailing frontal or conveyor belt structure. Warm spell structure follows a north-to-south gradient and peaks at high values, above 50%. This is not surprising since these storms occur mostly in warm seasons and in warm regions (see Fig. 7). Cluster 6 is particularly hot and is associated with a majority of heat lows over the desert (Givon et al., 2024). In clusters 7 and 5, the deflection of the near-surface warm spell frequencies associated with the storm is clearly visible, which outlines a possible role of Mediterranean cyclones in warm spells. Note that the warm spells are chosen here as exceeding the temperature above which heat effects on human health start to interact with dust concentration (Katsouyanni et al., 1993). They are not defined as a hazard in and of themselves, and should only be interpreted as such with some caution. For clusters 6 and 7, the compound PM10-heat footprints have a similar shape as the PM10 footprints, while it is displaced to the north in cluster 5. Note that the magnitude of the compound event probability is quite close to the PM10 probability in cluster 6, but much smaller for the two other clusters. This will be discussed in more detail in Sect.4.

## 4 Discussion

### 4.1 Relation between individual hazard and compounding frequencies.

In Figs.9-11, we noted that, for rain-wind and rain-wave events, compound event probabilities usually scale with the probabilities of both constituting individual hazard. This is to be expected if the different individual hazards involved co-vary and co-occur similarly across all clusters. Beyond such similarity across clusters, it becomes unclear how the temporal and structural characteristics of single hazards influence compounding. For example, wind footprint shape differences may be the reason why cluster 4, which is associated with higher probabilities of both rain and wind individual events than cluster 2, exhibits lower compound probabilities. Alternatively, clusters with very high PM10 probabilities frequently aren't associated with many warm spells and vice-versa. Such variations make it difficult to intuitively understand why certain clusters are associated with higher probability of compounding, and hence higher risk. To help understand what sets the probability of compound events in a given cluster, we start by quantifying the relation between the occurrence of individual events, and the occurrence

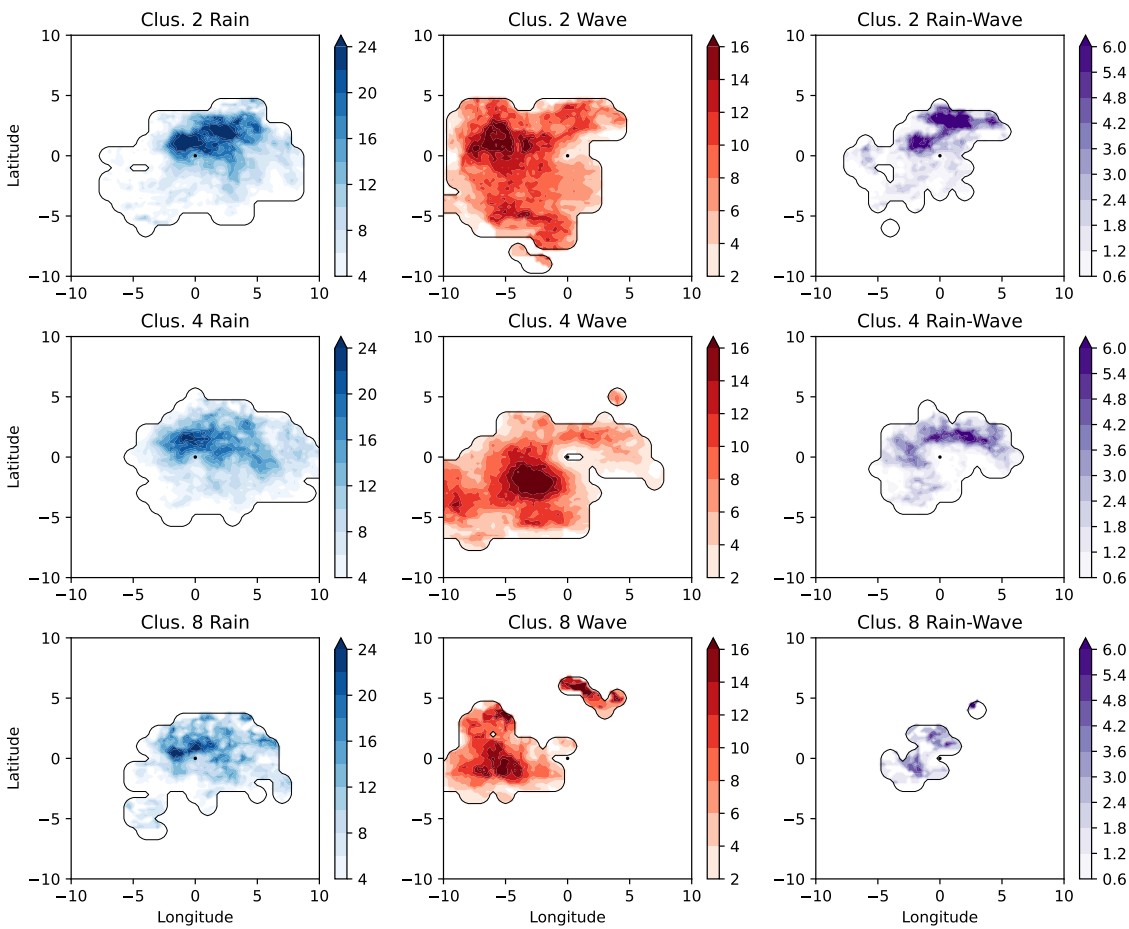

**Figure 10.** Storm-relative composite probabilities (shading) over the Mediterranean for rain events (blue, left column), wave events (red, center column) and compound rain-wave events (purple, right column) and for cluster 2 (top row), cluster 4 (middle row) and cluster 8 (bottom row). Probabilities are in %. The black contour identifies the statistically significant area for each hazard (see Eq.1).

of compound events. To do so, we introduce an "Ideal" compounding scenario, as well two co-occurrence metrics: "Simultaneity" and "Overlap". The Ideal scenario is a simple evaluation of the maximum compounding rate that could occur in a cluster if all the individual hazard events in that cluster were perfectly matched in space and across storms. Ideal compounding is defined as

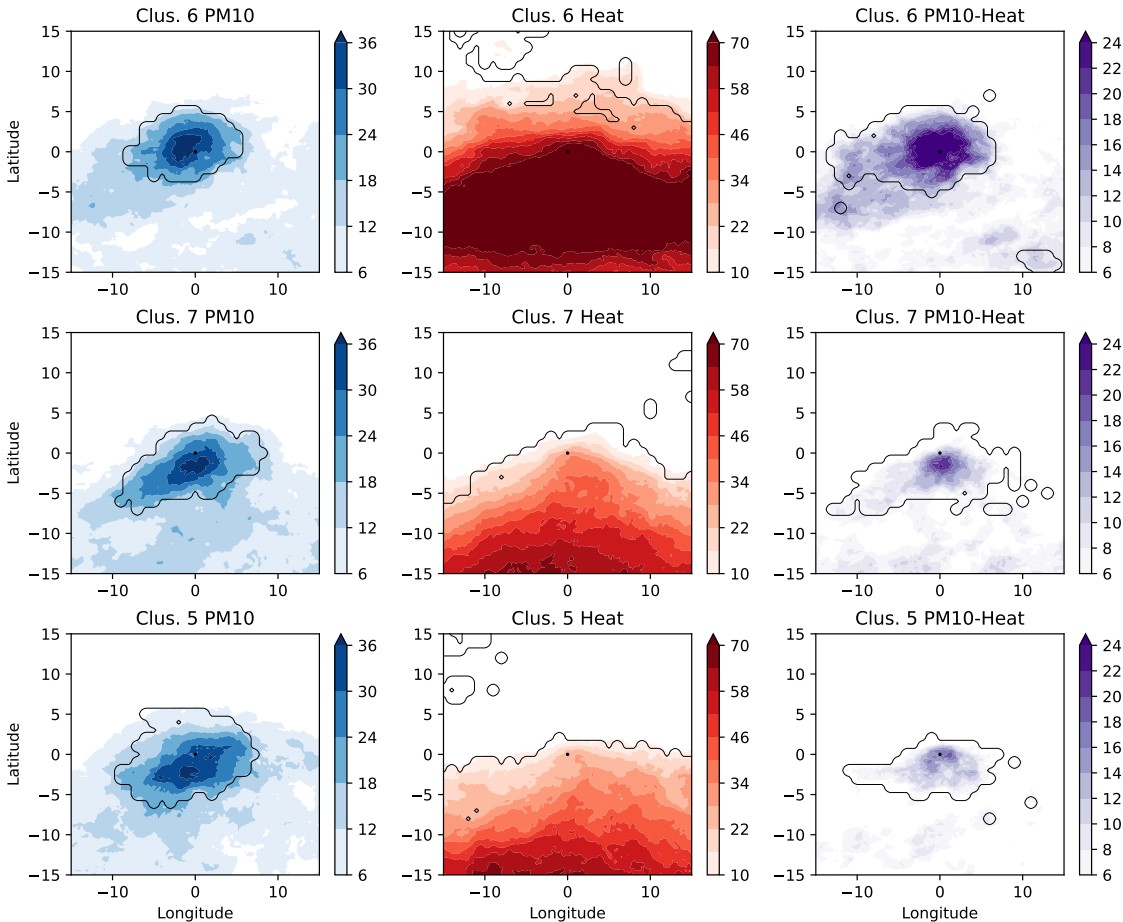

**Figure 11.** Storm-relative composite probabilities (shading) of PM10 events (blue, left column), warm spells (red, center column) and compound PM10-heat events (purple, right column) and for cluster 6 (top row), cluster 7 (middle row) and cluster 5 (bottom row). Probabilities are in %. The black contour identifies the statistically significant area for each hazard (see Eq.1).

$$P_I = \min(\overline{A}^{SH}, \overline{B}^{SH}), \tag{3}$$

where the overline denotes an average, and $S$ and $H$ are the storm event dimension and the horizontal dimensions respectively. As in Eq.2, the horizontal average is taken over a $20{\times}20$ degrees box. Using Eq.2, this formulation is equivalent to $P_I = \min(P_A, P_B)$. The left column of Fig. 12 shows the comparison between the ideal $P_I$ and the actual $P_C$ compound event

probabilities. The actual probability of events is much smaller than the ideal probability for both the rain-wind and rain-wave events, but not for the PM10-heat events. As suggested before, the ideal rain-wind compound probability of cluster 4 is larger than that of cluster 2 despite the actual probability being smaller. Similarly, the ideal PM10-heat probability of clusters 7 and 5 is higher than that of cluster 6 even though cluster 6 has the highest actual compound event probability. In fact, for cluster 6, the actual probability is almost as large as the ideal probability. To understand the disparities between ideal and actual compound

probabilities, we now turn to Simultaneity (Sim.), a measure of the extent to which different individual hazards tend to occur in the same cyclones within a cluster, and Overlap (Over.), a measure of how well individual hazard footprints correspond in space. Simultaneity and Overlap both vary between 0 and 1 and are given by

$$Sim = \frac{P_C}{\min(\overline{A}^S, \overline{B}^S)^H}, \tag{4}$$

$$Over = \frac{P_C}{\min(\overline{A}^H, \overline{B}^H)^S}, \tag{5}$$

where $P_C$ is defined as in Eq.2. The right column of Fig. 12 shows that Overlap is smaller in cluster 4 than in cluster 2, which finally explains the smaller rate of compounding in that cluster (Simultaneity is also smaller, but to a lesser extent). The relatively poor spatial match between the rain and wind regions in cluster 4 is the reason why it is not the cluster associated with the highest rain-wind risk. Similarly, the reason why PM10-heat events have probabilities of occurrence closer to $P_I$ than rain-wind and rain-wave events is that both Simultaneity and Overlap are higher. Very high values of Sim. and Over.,

around 80%, explain why cluster 6 exhibits the highest PM10-heat compound probabilities. This is due to the very high warm spell probability associated with heat lows over the desert in that cluster, which means that most of the time a PM10 event occurs, a warm spell is co-occurring. In general, clusters with higher compound event probabilities have higher single hazard Simultaneity and Overlap, with two interesting exceptions. First, Simultaneity is fairly constant in Rain-Wave events, which indicates that there are little variations across clusters of the distribution of individual hazard across cyclones within a cluster.

Second, cluster 9, which is associated with cut-off lows (Givon et al., 2024) has high Overlap and Simultaneity of events despite being associated with low probabilities of individual hazards. The latter may be due to the fact that cluster 9 has a broad geographical distribution, from the lee of the Alps to the Sahara (Givon et al., 2024), which may decrease the incidence of both individual and compound hazards without impacting Simultaneity or Overlap. Within that cluster, cyclones occurring over the desert may only be associated with warm spells and PM10, not rain, while the reverse may be true for one occurring over the

sea, but both may be associated with high Simultaneity and Overlap locally.

Multivariate compound events, which are the focus of this paper, require that the different hazards involved should occur at the same time and location, and these requirements determine the definition of Simultaneity and Overlap. The probability of other types of compound events, such as temporally compounding events where the risk increases due to hazards occurring in a

495 succession (Zscheischler et al., 2020), cannot be understood using these metrics. In addition, the hazards studied here have been

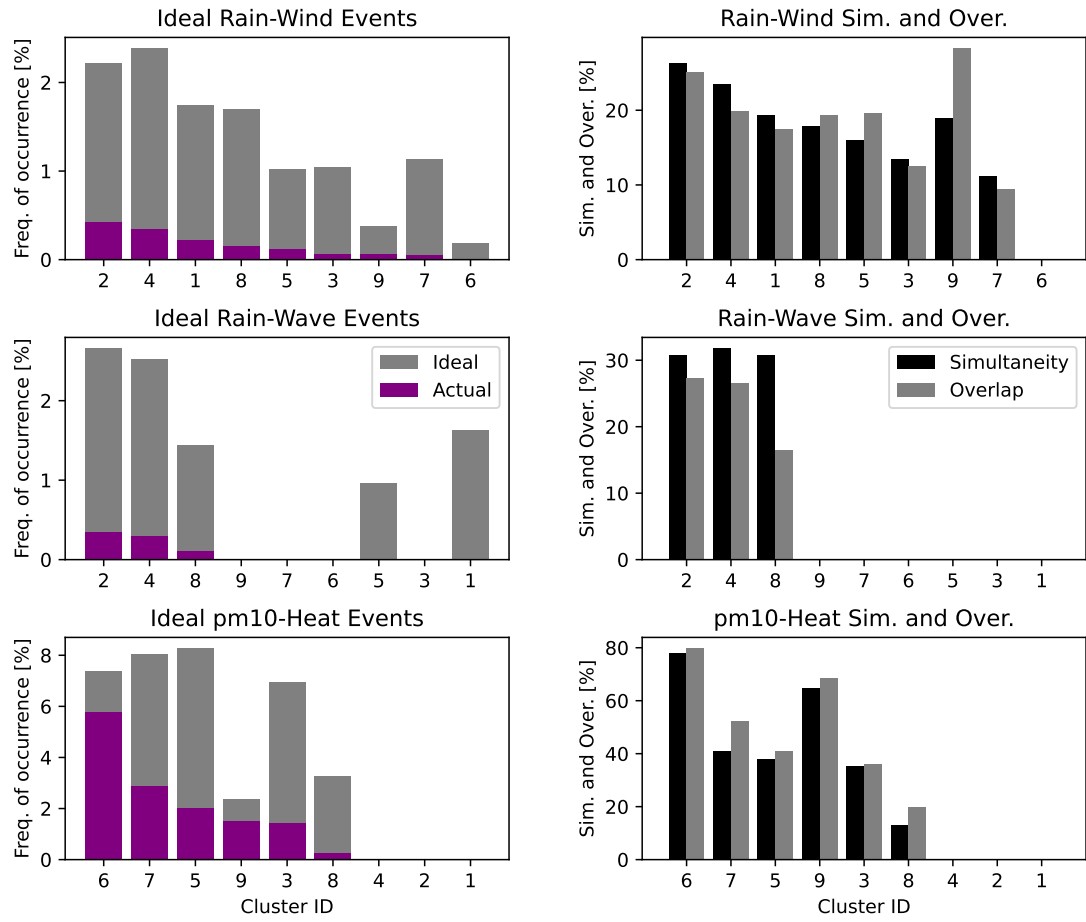

**Figure 12.** Left column: Ideal (grey bars) and Actual (purple bars) probabilities of occurrence of rain-wind (top), rain-wave (center) and PM10-heat (bottom) compound events. Right column: Simultaneity (black bars) and Overlap (grey bars) of individual hazards for rain-wind (top), rain-wave (center) and PM10-heat (bottom) compound events.

selected specifically because they lead to multivariate events and these hazards are not all prone to leading to increased impacts when occurring sequentially. For example the enhanced damage to infrastructure due to the rain and wind co-occurring, also called Wind-Driven Rain, depends on the rain falling at an angle because of strong winds occurring at the very same time (e.g., Blocken and Carmeliet, 2004). Wind occurring after rain, or vice-versa, would not lead to the same risk enhancement. Hence, 500 the future studies of different types of compound hazards will require the development of new metrics.

## 4.2 Topography

Overlap is particularly small for rain-wind events. This is consistent with the very distinct rain and wind footprints in Fig. 9, and may be due to the fact that cluster 2, 4 and 1 cyclones usually occur near the north shore of the Mediterranean, partly over water and partly over land. To illustrate the role of topography, Fig. 9 shows the 300 m isohypse along with a contour for the 100 m per degree meridional slope. Note that storms from all these clusters occur just south of the 300 m altitude average line and of a high average gradient area. The isohypse corresponds well with contours of high wind event probability. This is likely due to the fact that, over rough topography, the 10 m s$^{-1}$ lower wind threshold is higher than the 98th percentile of winds, and hence that wind events are, by our definition, less frequent in such areas. Precipitation tends to be more concentrated to the north of the cyclones, where it may be enhanced by the forced ascent over topography. Interestingly, topography is located closer to the average cyclone center in cluster 4, and extends further to the west. For that reason, wind events, and hence compound events are fewer to the northwest of the center, where they occur most frequently in cluster 2. Hence topography may cause a partition between rainy and windy areas in Mediterranean cyclones, and be responsible for the relatively small probability of rain-wind events in the region. Topography is not relevant to rain-wave compound events because only sea-level data points are considered, but it is worth noting that precipitation over the sea (Figs.5 and 10) has a similar magnitude as precipitation associated with all storms (Figs.4 and 9). This suggests that either topography does not enhance much the probability of rain events in Mediterranean cyclones, or that there is a compensating effect of moist air when storms occur over the sea.

## 4.3 Relation to dynamical features

We now turn to quantifying the relation between hazards and dynamical features such as warm conveyor belts, cold fronts and dry intrusions. To do so, we use the definitions introduced in Sect.2, and we compute the probability of co-occurrence of hazards with dynamical features. Figure 13 compares the probability of compound hazard with the joint probability of a compound hazard and a dynamical feature. First, note that the probabilities of co-occurrence with WCBs or with CFs are high and fairly similar for rain-wind and rain-wave events. This similarity of the magnitudes is certainly influenced by the definition of a 2.5 degrees region of influence around CF points introduced in Sect.2, but the good scaling between the CF and WCB probabilities is likely physical and due to the fact that both features tend to occur within the same cyclones. The good correspondence between these compound hazards and WCB or CF features suggests that the distinctive distribution of the hazards around the cyclone center (see Figs. 9 and 10) may be modulated by the dynamical features themselves. Then we see that dynamical features are associated with a far smaller fraction of PM10-heat events than of other compound events. This is likely because the cyclones associated with PM10-heat typically occur further south and in the summer, while the cyclones associated with rain, wind and waves are strongly baroclinic midlatitude winter cyclones. By contrast to their strong relation to WCBs and CFs, very few compound events co-occur with DIs. This result is partly due to the type of compound events considered; it makes sense that very few multivariate event involving precipitation would co-occur with dry intrusions, while DI occurrence is minimal during summer, when PM10-heat compound events peak. Other types of compound events not considered here, like waves and winds which poses a risk to navigation, would likely be more associated with dry intrusions. It also makes sense that

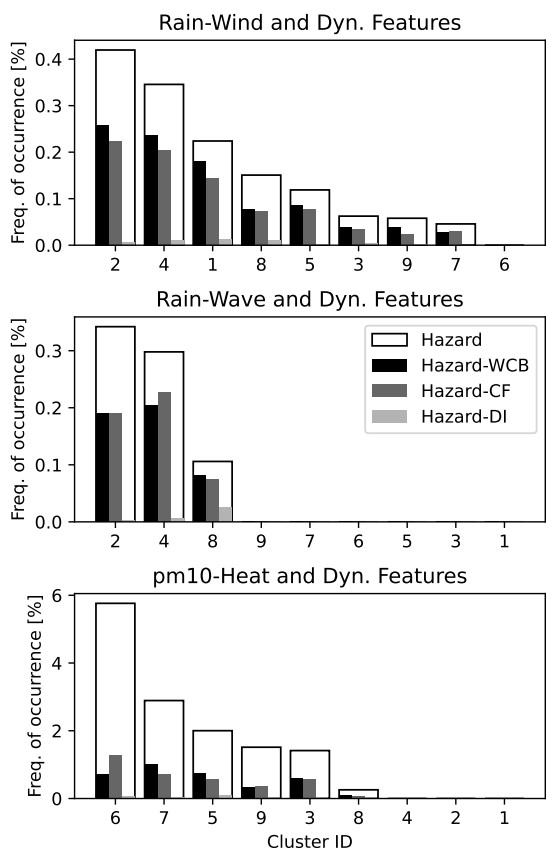

**Figure 13.** Probability of occurrence of rain-wind, rain-wave and PM10-heat compound hazards (broad white bars), and probability of co-occurrence of compound hazards with warm conveyor belts (WCB, black bars), cold fronts (CF, dark grey bars) and dry intrusions (DI, light grey bars). The clusters are sorted in descending order of hazard probability.

DIs would not be very important in the weakly baroclinic cyclones of cluster 6, 7 and 5. Interestingly, the fraction of compound events where dynamical features are involved in rain-wind or rain-wave events is somewhat lower for the highest-risk cluster 2. This may be an indication that, in clusters associated with the most events, the surface cyclone itself is associated with enough rain, wind and waves events to cause a significant fraction of the compound events. In slightly weaker cyclones, like those of clusters 1 (rain-wind) or 8 (rain-wave), it may be that the presence of a well-defined front or conveyor belt is often necessary for compound events to occur.

To refine our understanding of the role of dynamical features, we now quantify their relation to individual hazards. Figure 14 compares the probabilities of individual hazards with the joint probabilities of individual hazards and of dynamical features. Here as in Fig. 13, the hazard-WCB joint probabilities scale fairly well with the hazard-CF joint probabilities for all clusters and for a given hazard. From hazard to hazard, however, the relative magnitudes of WCB and CF co-occurrence probabilities vary substantially. For example, the joint probabilities of rain events with WCBs are always larger than with CFs, while on the contrary, the joint probabilities of wind or wave events with WCBs are always smaller than with CFs. We interpret these results to mean that, while CFs and WCBs tend to occur within the same cyclones, WCBs are more important in controlling precipitation in Mediterranean cyclones, while CFs are more important for wind and wave events. As a result, for the compound events considered in Fig. 13, which require the joint occurrence of both rain and wind, or rain and wave events, the relative roles of WCBs and CFs are more balanced.

Almost no rain events are associated with dry intrusions, which explains the lack of association of that feature with compound events involving rain. Dry intrusions have small but non-zero joint probabilities with wind, waves or PM10, and wave events are the only ones where DIs become more important than other dynamical features (clusters 8 and 5). DIs tend to occur far from the storm centers, in storm-relative regions where the hazards are more diffuse and rarely statistically significant. The distance between most DI occurrences and the storm center differs from the conceptual picture presented by Browning (1997), which is based on Atlantic storms and may reflect a weaker association between these dynamical features and Mediterranean cyclones. In addition, the fact that we are only accounting for hazards in statistically significant areas here may explain the relatively small role of dry intrusions with respect to other dynamical features. It is likely that accounting directly for the relation between features and hazards without taking storm relative composites would reveal a larger role for dry intrusions (e.g., Fluck and Raveh-Rubin, 2023; Nissenbaum et al., 2023), but such an analysis exceeds the scope of this climatology. One example of a climatology revealing a role for dry intrusions in compound events is the study of Portal et al. (2024), which shows, in an Eulerian perspective, an association with wave and wind compound events in the Mediterranean. Overall, dynamical features play the largest role in rain events, where they are associated with more than half of all events, a smaller role in wind and wave events, where they are associated with a third to half of the events, and an even smaller role in PM10 events. Note that the cluster associated with the highest probability of PM10 events is cluster 4, for which WCBs, CFs and DIs all play a role, albeit not very large. While it isn't associated with any significant probability of PM10-heat events, given the relatively high latitude at which cyclones occur in that cluster (see Fig. 7), and its strong association with PM10 events, it is likely that the Type B frontal cyclones of cluster 4 are major contributors in transporting dust from the Sahara to Europe.

### 4.4 Comparison with conditions in the absence of storms.

In this paper, we have focused on the frequency of compound hazard within storms, but not on the general occurrence of compound hazards in the region. This endeavor is outside the scope of this paper and is addressed for Rain-Wind events in (Portal et al., 2024). Taking an Eulerian perspective over the Mediterranean Basin, they show that locally, Rain-Wind event frequency is enhanced by a factor of at least 5 (and sometimes much more) by the presence of a cyclone. In the present study,

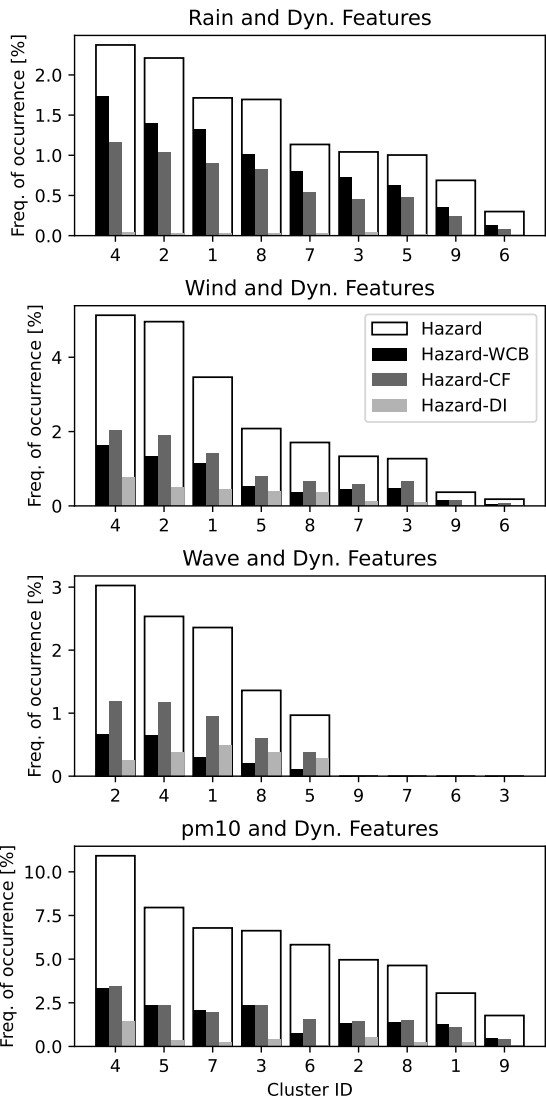

**Figure 14.** Probability of occurrence of rain, wind, wave and PM10 individual hazards (broad white bars), and probability of co-occurrence of individual hazards with warm conveyor belts (WCB, black bars), cold fronts (CF, dark grey bars) and dry intrusions (DI, light grey bars). The clusters are sorted in descending order of hazard probability.

examining storm-relative composites (Figs. 9, 10 and 11) provides some insight on the increase of hazard likelihood brought about by the presence of the storms, in comparison to that at the same locations and for seasonally similar times. The relative difference between the maximum and the minimum event probabilities within the statistically significant region gives a sense of the influence of the storm with respect to background value. For example, there is roughly a factor 10 between the probability minima and maxima within the statistically significant regions in Fig. 9, and those minima still represent probabilities that are significantly higher than the background. There is somewhat less difference between the maxima and minima for the rain-wave events in Fig. 10, due to the wave background values being high. The relative difference between maxima and minima is yet again smaller for PM10 events, and hence PM10-heat events (Fig. 11), because the cyclones in these clusters occur over the Sahara (see Fig. 7), a part of the world where dust emissions are particularly high.

## 5    Conclusions

Mediterranean cyclones, despite their relatively small sizes and intensities, are associated with much of the weather-induced damage in the region (Flaounas et al., 2015, 2022). While our fundamental understanding of Mediterranean cyclones, has progressed rapidly, our understanding of the associated hazards and impacts remains somewhat limited, owing to the complexity of the Mediterranean basin and the recency of extensive research on the topic (Flaounas et al., 2022). For example, while the relation between many individual hazards and Mediterranean cyclones is already fairly well understood, the relation between Mediterranean cyclones and the potentially more impactful multivariate hazards (Zscheischler et al., 2018) remains to be characterized. To fill this knowledge gap, we aimed to establish a climatology of the compound hazards that occur in association with Mediterranean cyclones. To that effect, we devised a simple method to compute storm-relative composites of multivariate compound events, and evaluate their statistical significance. Composites were computed for rain-wind, rain-wave (which required evaluating land-masked composites) and PM10-heat events. These three types of multivariate events were selected because they consider multiple individual hazards and a broad range of societal impacts, from infrastructure damage to human health. To help assess the role of the large-scale environment, the climatology is based on a state-of-the-art cyclone tracks dataset for the region (Flaounas et al., 2023) and a classification of those tracks based on the upper-level PV field associated with the cyclones (Givon et al., 2024). In addition, to assess the role of dynamical features, we evaluated the probabilities of co-occurrence of the compound events and of warm conveyor belts, cold fronts and dry intrusions. To summarize, this study established multivariate hazard footprints relative to Mediterranean cyclones for various large scale configurations and in relation to cyclone dynamical features, aiming to provide information on cyclone-relative hazard occurrence, and on the contribution of various scales and processes to that occurrence.

We showed that distinct large-scale contexts are associated with different types of compound events in Mediterranean cyclones. Namely, rain-wind and rain-wave events are mostly due to type A and B cyclones as well as Rossby wave breaking near the Mediterranean north shore in cold seasons, while PM10-heat events are due to heat lows, daughter cyclones and anticyclonic wave breaking in North-Africa, predominantly during transition and warm seasons. Event probability footprints varies between

storm clusters but much more so between hazards. In general, wind events are concentrated to the south of the storms, rain events are concentrated to the north, wave events are concentrated to the west, and dust events are concentrated either around the storm center or to the south. The compound hazard footprint shapes then depend upon the relative positions of the individual hazard footprints, with rain-wind event probabilities peaking west-north-west of the cyclone centers, rain-wave events peaking to the north, and PM10-heat events being distributed around the storm centers. We find that the probabilities of compound events depend mainly on the probabilities of individual hazards, but also on how well these hazards coincide temporally and spatially. To understand how high compounding probabilities arise, we introduce the Simultaneity and Overlap parameters which reveal that in general, individual hazards coincide at a higher rate in clusters associated with high compounding probabilities. For example, the anticylonic wave breaking cyclones of cluster 2 are associated with the most rain-wind events because the individual hazard footprints overlap particularly well. Finally, we quantified the relation between dynamical features and compound events, and we found that warm conveyor belts and cold fronts were both equally important for rain-wind and rain-wave events, while no dynamical feature was critical to the occurrence of PM10-heat events. Dynamical features were shown to co-occur more frequently with compound events in clusters associated with a smaller probability of compound events.

We hope that the information provided here will help interpret the risk posed by different weather systems in the Mediterranean, and infer how that risk may change in the future. For example, the observation that the frequency of cluster 6 heat lows has been increasing in recent decades (Givon et al., 2024) suggests that an increase in PM10 and warm spells is probably currently occurring and may be expected to continue. Next steps in the present research endeavour will include establishing an Eulerian climatology of compound events in association with cyclones in the Mediterranean region.

*Code availability.* The code used to produce analyses is available upon request.

**Appendix A: Supplementary figure**

*Author contributions.* Raphaël Rousseau-Rizzi, Olivia Martius, Shira Raveh-Rubin and Jennifer Catto designed the study. Raphaël Rousseau-Rizzi and Alice Portal developed the analysis code and performed the analyses. Raphaël Rousseau-Rizzi prepared the manuscript with contributions from all co-authors.

*Competing interests.* At least one of the (co-)authors is a member of the editorial board of Weather and Climate Dynamics.

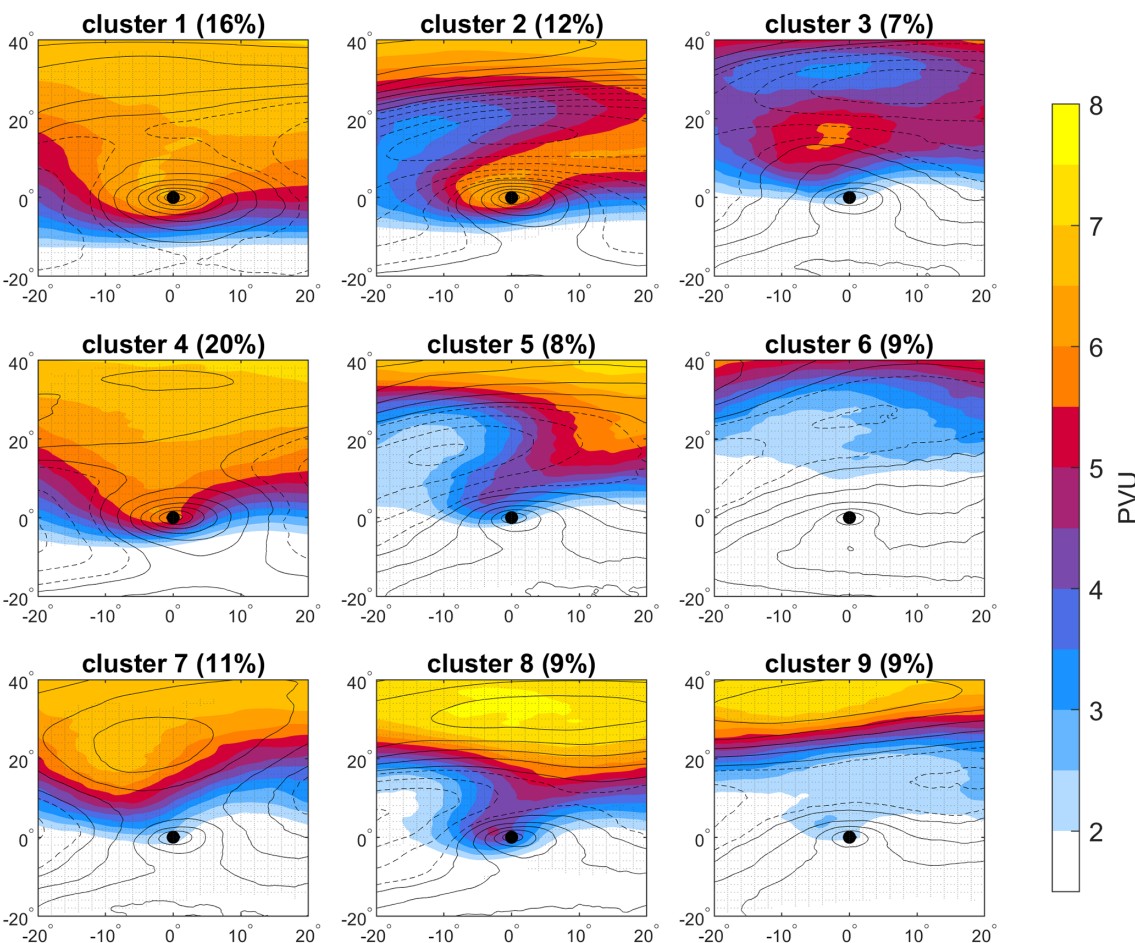

**Figure A1.** Large scale PV environment of cyclone clusters reproduced from Fig. 3 of (Givon et al., 2024). Original caption: "*Cyclone-centered cluster composites of upper-level (320–340 K averaged) PV (PVU, shading) and SLP (black contours at 2 hPa intervals, dashed above 1015 hPa). Stippling indicates a 99% significance level of the PV field concerning the total cyclone average (Fig. 2a). The mean frequency of each cluster out of all cyclones considered is given in the title.*" Clusters 2, 4 and 1 are associated with Rain-Wind events, Clusters 2, 4 and 8 with Rain-Wave events, and cluster 6, 7 and 5 with PM10-Heat events.

*Acknowledgements.* The authors would like to thank the Swiss National Science foundation for supporting this work under grant number IZCOZ0 205461. SRR and YG acknowledge funding from the Israel Science Foundation (grant number 1242/23) and the De Botton Center for Marine Science at the Weizmann Institute. The authors would also like to thank the MedCyclones COST action for facilitating meetings on the topic of this study, and the participants to the meetings, for stimulating discussions and insight. Finally, the authors would like to thank Katharina Heitmann and Michael Sprenger for providing the warm conveyor belt data used in this study.

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
