# Peer review of "A storm-relative climatology of compound hazards in Mediterranean cyclones"

_EGUsphere, 2023_

## Referee Comment (RC2)

**Review of egusphere-2023-2322 – Rousseau-Rizzi et al. – A storm-relative climatology of compound hazards in Mediterranean cyclones**

The manuscript by Rousseau-Rizzi et al. discusses a system-relative climatology of compound events associated with Mediterranean cyclones focusing on wind-rain, rain-waves, and particulate matter-heat events. The study shows that compound events often happen during specific clusters, which is also helpful knowledge for forecasting future events. Overall, the work shows great value and is well written. However, I have two concerns discussed below that I think need attention before publishing.

**Main concerns**

1. **Only using one time step per cyclone**

   I understand that only using one time step simplifies this work tremendously. However, some of these hazards and dynamical features discussed occur during different times of a cyclone's life cycle (e.g., Hewson and Neu (2015) and more recently Eisenstein et al. (2023), although these works have been done over the North Atlantic and Central Europe). I see problems only using the time of minimum pressure as it will exclude a lot of information and hazards neglecting the development of a storm. Maybe some compound events happen sooner or later in the life cycle? Even though I understand how much work it would be to include more time steps, especially considering possible double-counting of cyclones as you mentioned, I believe this needs to be included to make this work even more meaningful. At least a detailed discussion about what would change, looking at other time steps with a few examples, must be added. Also, how important are hazards that are detected immediately after one another?

2. **Dynamical features**

   I am missing a clear description of the dynamical features, their hazards (wind, precipitation), etc. There are numerous studies of the features in extratropical cyclones (e.g., regarding wind: Hewson and Neu, 2015; Clark and Gray, 2018; Eisenstein et al., 2022,2023; regarding precipitation: Catto and Pfahl, 2013; Catto, 2016, …). Do they differ in the Mediterranean/how so?
   For example, regarding the WCB: Clarify whether you are only interested in winds (warm conveyor belt jet; see literature above) or also in the forming cloud head which is responsible for most of the precipitation (Catto, 2016), which I assume as you consider it up to a height of 400hPa. Maybe consider explaining the difference of the warm conveyor belt jet, responsible for the high surface winds, and the WCB forming the cloud head and precipitation.
   I believe a more in-depth explanation and discussion is needed, especially for readers less familiar with these dynamical features.

**General comment:**

- You include datasets based on ERA-Interim and ERA5. Please clarify what inconsistencies -if any- might occur in doing so.

**Minor comments:**

*Abstract*

- l. 5f Please be clearer about what kind of classification and the "few different large scale configurations".

*Introduction*

- l. 24 and later: You introduce pm10 here for particulate matter. Later (l. 117) you introduce it for particulate matter specifically of size 10μm. Please clarify this already in the beginning.
- You consider both winds and waves as hazards, so I would suggest including a quick explanation how both influence each other, see e.g., Gentile et al. (2021) and Gentile and Gray (2023; also including dynamical features)
- l. 44f: Please refer to these studies directly instead of a not-published thesis.
- l. 79f: Please introduce the abbreviations of all features in the same paragraph (→ l. 39) and then use these abbreviations throughout the manuscript.

*Data and Methods*

- l. 101 Which figures to you refer to by Fig. "X and Y"? I had a hard time following the different clusters. Please include at least a small description with characteristics and a similar figure to Givon et al. (2023) of the clusters in the appendix or supplementary material. Maybe also show and discuss Fig. 8 earlier. This would make it easier for the reader to follow your discussion.
- l. 114 This sentence is confusing to me. Do you mean you take the average height of the top 33% wave heights? Please clarify and rephrase this sentence.
- l.128 What do you mean by distinct? A few more details about the detection methods should be added for understanding.
- l. 132 Is there more work on this than a not yet published thesis that can already be cited here?
- Fig. 2 Consider swapping the subfigures to be consistent with the order of how they have been introduced. Also, add (a) and (b).
- l. 161: "30°C"
- l. 172f See main comment 2:  Which hazards associated with fronts are you interested in – wind, precipitation, or both? Describe the hazards associated with the features.
- Is Cyclone ID = Storm ID?
- l. 206f I would suggest adding a half sentence explaining Monte-Carlo samples and/or include a reference.

*Results*

- l. 227 "have a points"
- l. 240 and throughout the chapter: Use Fig. X and Sect. X.
- l. 266 What do you mean with "cluster 2,4?" ?
- l. 297 Again, consider putting figures like this that you do not deem important enough for the main manuscript in the Appendix/Supplementary material.
- l. 303 I suppose these are some remnants of an earlier version/comments?

- l. 328 This seems to be similar to Central Europe then (Hewson and Neu, 2015; Eisenstein et al., 2023). Did you expect differences?
- l. 332 "has and intense"
- l. 335 and 228 Is this the cold conveyor belt (jet)? Or the precipitation associated with the cloud head/WCB? Please comment on this.
- Fig. 9,10,11 Consider using "row" instead of "line". Further, I suggest changing the caption to "Probabilities (shading) […]."

*Discussion*

- How would events be taken into account that are detected shortly after one another (if you would look at more time steps, main comment 1)?
- l. 399 "shows that_ overlap […]"
- l. 452ff the reader would benefit from a clearer description of the WCB earlier here (see main comment 2)
- Same paragraph: What about turbulence of the features, affected area etc.
- l. 463 How come DIs tend to occur far from the cyclone centre comparing it to the figures of Browning (1997)?

*Conclusion*

- l. 506 "where"?

---

## Author Comment (AC1)

**Response to Reviewers: "A storm-relative climatology of compound hazards in Mediterranean cyclones"**

Raphaël Rousseau-Rizzi, Shira Raveh-Rubin, Jennifer Catto, Alice Portal, Yonatan Givon and Olivia Martius

**Reviewer 1**

We thank the reviewers for their very insightful and valuable comments, and for all of their work in helping to improve this paper. We hope that the improvements to the paper will correspond to the reviewers vision and expectations.

The study by Rousseau-Rizzi and collaborators sought to identify and quantify the coexistence of two hazards related to cyclonic activities over the Mediterranean Sea. The analysis was based on three pairs of hazards: rain and wind, rain and waves, and concentrations of PM10 and heat. The outstanding contribution of the study lies in its separate analyses for different types of cyclones, with clusters classified by upper-air potential vorticity. Another interesting investigation concerns the probability of occurrence of each pair of hazards in situations of warm conveyor belts, cold fronts and dry intrusions, always associated with cyclone's passages. Perhaps the most delicate point concerns selecting hazard events based only on one hour of the life cycle of a Mediterranean cyclone, significantly limiting the identification of potential extreme risks if we consider that each risk has its maximum at different moments of a storm. In any case, the study is presented coherently and in detail, and its publication makes essential contributions to quantifying hazards associated with Mediterranean cyclones.

Please find below a list of some questions, suggestions and corrections regarding the manuscript:

[ Lines 23-24 ] "a broad variety of hazards ranging from sea waves"

[ Suggestion ] Perhaps this list could be extended to storm surges, coastal erosion, floods and landslides, as the passage of cyclones is also directly associated with these events.

[ Response ] Extending the list is a good idea. Since the hazards mentioned by the reviewer are mostly modulated by the rain and wave hazards that have already been mentioned, we have taken the comment into account by stating in the manuscript: "Precipitation and waves for example, may impact society via flooding, coastal erosion and landslides.". For simplicity, and because they are not a dominant hazard in the Mediterranean, we are not mentioning the storm surge in the paper.

[ Line 24 ] "pm10"

[ Correction ] In the related literature, pm10 is spelled PM10. Please replace this in the following appearances.

[ Response ] All instances of "pm10" have been replaced by "PM10" in the text

[ Lines 44-45] "citation of currently embargoed thesis by Laura Owen"

[ Correction ] Unpublished studies cannot be cited. Everything written in this manuscript will be further cited referring to this study. Consider removing this information or inviting Laura Owen as a co-author if it is relevant to a future Owen publication. This should also be applied to the following citations related to Owen's work.

[ Response ] Fortunately, the embargo was lifted in February 2024, and the publication is now properly cited.

[ Line 49 ] "PV"

[ Correction ] Please move the acronym's introduction from line 96 to here on its first occurrence.

[ Response ] Thank you for noticing the error. It has been corrected.

[ Line 64 ] "... on three types of multivariate compound events,"

[ Question ] Why not wind and wave?

[Response] The paper is becoming quite lengthy and we wanted to keep it approachable. We have added in the text: "For brevity, wind-wave events are not analyzed in this paper."

[ Line 65 ] "heat waves"

[ Correction ] This hazard was not assessed in this study. Perhaps warm-spell is more appropriate here.

[Response] We thank the reviewer for this observation. The terminology has been adjusted.

[ Line 70 ] "cold fronts (CF)"

[ Correction ] To make the acronyms' identification more accessible, I suggest moving the introduction of CF to line 39, where WCB and DI are also introduced.

[ Response ] Thank you for the comment. The introduction of CF has been moved accordingly.

[ Line 77 ] "Data and methods"

[ Question ] Could you clarify which dataset was used to define the study period, as several datasets covering different periods were utilized?

[Response] Thank you for the comment, the period used was determined by the largest overlap of the datasets available. Hence, we specified "Since all datasets, except PM10, are available for the period 1980-2019, all analyses cover that period, except for PM10-related analyses, which extend from 2004 to 2019."

[ Line 90 ] "ERA5 reanalysis data"

[ Correction ] You must cite Hersbach et al (2020) instead of Flaounas et al (2023).

[Response] Thank you for the comment, this has been done.

[ Line 91 ] "leads to accurate tracking of storm centers"

[ Suggestion ] In this case, I would be more accurate by adding " at the Mediterranean scale" to this sentence.

[Response] Thank you for the comment, this has been done.

[ Line 92-93 ] "... activity around Italy and southern France as well as over the Sahara and south of Turkey."

[ Suggestion ] In Mediterranean cyclone studies, this region is often known as "the lee of the Alps", referring Buzzi & Tibaldi (1978). The same works for the Sahara and Turkey. I suggest referring to the Atlas and the Anatolian mountain chains instead of associating these positions with country or region names.

Buzzi A, & Tibaldi S (1978). Cyclogenesis in the lee of the Alps: a case study. QJR Meteorol Soc 104:271–287. https://doi.org/10.1002/qj.49710444004

[Response] We agree with the comment. The text now reads "The figure outlines hotspots of storm activity in the lee of the Alps (as identified by Buzzi and Tibaldi(1978)) as well as over the Sahara, in the lee of the Atlas, and south of the Anatolian mountains."

[ Line 93-94 ] "... there is more storm activity over bodies"

[ Suggestion ] Depending on the tracking method, more storms can be identified over land during the summer months. As Flaounas et al (2023) presented composite tracks, only those simultaneously identified by N methods were considered (in this study, N is also called Confidence Level), keeping only parts of tracks in which the different methods overlapped. This explains the high number of tracks over sea once there are divergences between methods in identifying tracking points over land. Perhaps you should add a comment pointing to this.

[Response] Very good point, thank you. We have added "The method used by Flaounas et al. (2023) to produce composite Mediterranean cyclone tracks requires the agreement of multiple (here chose to be 5) different tracking methods. The fact that these methods tend to agree better over the sea than over land (Flaounas et al., 2023) partially explains the

higher track density over bodies of water.

[ Line 97-98 ] "... the upperlevel potential vorticity (PV)"

[ Suggestion ] Considering the importance of this clustering for constructing your analyses, it would be interesting to describe them briefly and explain why you chose so many clusters (9).

[Response] We agree that more details are beneficial early on, hence we have added a description of the clusters in Sect. 2.1. The cluster definition is not introduced in this paper but in Givon et al. 2023. The new description clarifies this and, by detailing the differences between clusters, identifies why so many clusters are required.

[ Line 98] "... introduced by Givon et al. (2023)."

[ Suggestion ] This method dates back to the 90s and early 2000s with several works from K. Emanuel, R. Romero, and others. It is OK if you used Givon et al (2023), but I think it is essential to cite works such as:

Davis K A & Emanuel K A (1991). Potential Vorticity Diagnostics of Cyclogenesis. Mon. Wea. Rev., 119, 1929–1953.

https://doi.org/10.1175/1520-0493(1991)119¡1929:PVDOC¿2.0.CO;2

Romero R (2001). Sensitivity of a heavy-rain-producing western Mediterranean cyclone to embedded potential-vorticity

anomalies. Q.J.R. Meteorol. Soc., 127: 2559-2597. https://doi.org/10.1002/qj.49712757805

[Response] We think that in context, the following text make it clear that this specific classification using SOM was introduced by Givon et al. (2023): "[...] classification of Mediterranean cyclones introduced by Givon et al (2023). In this classification, the cyclones are organized into nine clusters based on the surrounding PV field averaged over the 320-to-340 K dry isentropic levels at the time of minimum central pressure, and using self-organizing maps [...]" . We do not claim that potential vorticity diagnostics were introduced in this work and would prefer not to modify the text here.

[ Figure 1 ] ...

[ Correction ] Name the colour scale units and, if possible, enlarge the whole figure.

[Response] The figure has been enlarged and we now specify "in units of number of cyclones per year per 1×1 degree box".

[ Line 100 ] "... using self-organizing maps"

[ Correction ] Self-Organizing Maps (capitalize each word regarding acronyms).

[Response] Thank you for the comment, this has been done.

[ Line 101 ] "... as the associated mean PV fields can be viewed in Figs. X and Y"

[ Correction ] I think something is missing here.

[Response] Thank you for the comment, the figure numbers were indeed missing and have been substituted in for the X and Y.

[ Lines 102-103 ] "... only the large scale environment of storm clusters shown to be responsible for compound events will be discussed."

[ Question ] Is this "the large scale environment of storm clusters" defined by Givon et al (2023), or is a subsequent filter applied to keep only these storms?

[Response] Yes, that is the case. since the whole discussion of the PV fields has been modified, this sentence has been repurposed and is now: "In the present study, only the storm clusters shown to be responsible for compound events will be discussed in detail"

[ Line 106 ] "... heat wave data"

[ Correction ] ERA5 ReAnalysis does not include heatwave data. More than that, this study does not address the heatwave phenomenon. Please cite air temperature.

[Response] Thank you for the comment, this has been corrected.

[ Line 107 ] "... but only saved at 6h intervals"

[ Correction ] What about the spatial resolution?

[Response] Thank you for the comment, the resolution (0.5 degrees) is now specified in the text.

[ Line 118 ] "... (CAMS) reanalysis"

[ Correction ] Please, add the dataset spatial-temporal resolutions.

Thank you for the comment, we have added " at a resolution of 0.75 degrees"

[ Lines 127-128 ] "Note that the methods used to identify DIs and WCBs are similar, while the front identification method is very distinct."

[ Suggestion ] Better understanding of this information is essential. Consider adding short sentences summarizing these methods.

[Response] We agree and have added more detail: "The DI identification is based on a systematic airmass trajectory computation, where DI trajectories are defined by their descending of at least 400 hPa within 48 h. A DI is then considered present at a given grid

point and time if any DI trajectory is present there at a pressure level of 700hPa or higher (i.e., in the lower troposphere). Thus, we consider so-called "DI outflows" as referred to in Catto and Raveh-Rubin (2019) and adapted here to ERA5. Similarly, the WCB identification is based on trajectories that ascend 600 hPa or more within 48 h. The density WCB trajectories is then computed for three different pressure intervals (Heitmann et al., 2023): an inflow interval (between the surface and 800 hPa), an ascent interval (between 800 and 400 hPa), and an outflow interval (pressures lower than 400 hPa). While the methods used to identify DIs and WCBs are similar, the front identification method (intoduced by Berry et al., 2011) is very distinct. The method first identifies points where the gradient of the thermal front parameter (Renard and Clarke, 1965) is zero, and then connects those points to form 1D features. In the method of Catto and Pfahl (2013), cold front are then identified based on their direction.".

[ Lines 135-136 ] "a given hazard magnitude may be benign in one region and catastrophic in another, and local percentile thresholds are most appropriate to capture the local risk. Otherwise, a fixed threshold may be used to define the occurrence of events everywhere."

[ Suggestion ] Check if a citation is needed. There is something in this sentence that looks like a déjà vu.

[Response] To make sure, we have googled the text and did not find any reference with text close to the statement in this sentence, other than the manuscript under review. The manuscript was also written entirely without the help of LLMs. This makes us confident in the relative originality of the text extract (we say "relative" because we do not claim to have discovered that the use of percentiles was an appropriate way to evaluate risk...).

[ Line 139 ] "... 99th percentile"

[ Question ] Have you computed the 99th percentile considering all hourly data, or did you filter hours without precipitation (¡ 1 mm/h)?

Hours without precipitation often accumulate 70-90% of the distribution. Including this in percentile computation can mask the actual threshold.

[Response] We have considered all data because we wanted the percentiles to represent similarly rare events in all locations, except where there is almost no rain. Excluding hours without precipitation in the calculation wouldn't allow for this comparison of similarly rare events which is, as stated in the text, useful to represent local risk.

[ Line 148 ] "... and interestingly, wherever Mediterranean cyclone track density is high"

[ Suggestion ] From my point-of-view, it is expected since the cyclogenesis and cyclolysis hotspots in the Mediterranean regions along the mountain chains. Try to rewrite and add this information.

[Response] Our statement was meant to be general to the map of computed thresholds, which extended outside of the Mediterranean basin. Similar features which have nothing to do with the Mediterranean Cyclones could be seen outside of the basin, for example in the Norwegian Alps. To account for this yet to make the text less confusing, the sentence ending was rewritten: "and, within the Mediterranean Basin, wherever Mediterranean cyclone track density is high".

[ Line 162 ] "30 C"

[ Correction ] 30°C?

[Response] Thank you for the comment, this has been corrected.

[ Lines 161-162 ] "Finally, for heat events, we chose a fixed 30 C threshold to be consistent with previous studies of increased mortality in the

event of co-occurring dust and heat extremes (Katsouyanni et al., 1993)."

[ Correction ] Divergence. Until now, the authors introduced a heat component described by heatwave events. But here, the threshold definition seems more similar to a warm-spell description and changes the final number of events a lot. Please note that the previous information regarding this component must be adjusted.

[Response] Thank you for the insightful comment, the terminology of the entire manuscript has been adjusted. For example, "heat events" has been replaced with "warm spells".

[ Line 176 ] "... only at the time of minimum central pressure."

[ Question ] How much of the total storm damage do the authors estimate to cover using a single time to represent a storm track as a whole? Consider that until here, it is not clear what the timestep used (1h or 6h). Also, consider that during a storm life-cycle, there are several moments of severe weather (precipitation, wind, waves, etc), and often it doesn't happen at the same time as the minimum central pressure.

Thank you for the insightful comment. We have clarified that the timestep used is 6 hours, due to limitations in ERA5 data storage capacity.

Since we want each storm to contribute equally to the composite (we want the composites to represent an average storm), we do not attempt to represent the whole storm track. However, we do want to produce a composite that is representative of the magnitude of the hazards encountered on average in a given storm cluster. Hence, it is important to verify that the time of minimum central pressure is such a representative time.

We have attempted to address this important point thoroughly while remaining within the scope of the paper. Once the clusters of interest are identified (2-4-1 for Wind-Rain, 2-4-8 for

Wave-Rain and 6-7-5 for PM10-Heat), we present a figure which shows the time evolution, 6 hourly, of the composite multivariate hazard likelihood, centered on the time of minimum central pressure. This new figure (Fig. 8, included at the end of this document) shows that the time of minimum pressure is associated with the maximum compound hazard likelihood in most clusters. Even in the clusters where another time is associated with higher hazards, the difference is small and we think the time of minimum pressure is still representative of the hazard maximum (described in Sect. 3.3). Hence, to make sure that the composites remain easy to interpret, we keep on using the time of minimum central pressure as a unique compositing time step, but that choice is now better justified.

[ Lines 176-178] "Since the time resolution of the storm tracks is sometimes higher than that of the hazard data, we select the minimum pressure over the times when hazard data is available."

[ Correction ] It is impossible, considering the information the authors have given up to this point (hourly hazard data X 6-hourly cyclone tracks data).

[Response] There seems to be confusion here, perhaps due to the manuscript. The hazard data is 6-hourly, while the track data is available hourly. This has now been clarified in the text.

[ Line 183 ] "... Storm ID"

[ Question ] It is not clear what Storm ID is.

[Response] This could indeed have been clearer. To clarify, we have added the sentence "Each individual storm matrix is attributed an index (Storm ID).".

[ Line 183 ] "... the Cyclone ID dimension."

[ Question ] What is Cyclone ID dimension?

[Response] We meant "Storm ID dimension", which has now been substituted in the text.

[ Line 196 ] "... over the Mediterranean"

[ Correction ] Do you mean, Mediterranean Sea?

[Response] That is correct. We have amended the text accordingly.

[ Line 233 ] "... the horizontal (latitude-longitude) dimensions of a 20×20 degree box around the cluster center."

[ Question ] As the search box is always a 20x20 degree area, can we consider these "horizontal dimensions" a unique and constant value for all storms?

[Response] The latitudinal dimension is unique and constant and the longitudinal one isn't.

This choice is in keeping with other literature on the topic including the classification of Givon et al. (2023). We have now added a statement to clarify this in the text:

"We also note that the choice of cyclone-relative latitude and longitude coordinates means that the size of the area changes between events depending on the latitudinal position of the cyclones. This approach is chosen to ensure comparability with other studies (e.g., Givon et al. 2023)."

[ Line 234 ] "... which may be interpreted as the fraction of time"

[ Correction ] In this case, it is not a matter of time but of cases, so it must be a "fraction of all cases".

[Response] Thank you for the comment, this has been corrected.

[ Lines 246-247 ] "This leads us to the observation that rain-wind compound probabilities are systematically multiple times smaller than either rain probabilities or wind probabilities alone."

[ Question ] Wasn't this expected?

[Response] In a sense, yes, but it had not been verified before for the Mediterranean Cyclones.

[ Lines 249-250 ] "Notably, cluster 2 which exhibits the highest compounding probabilities, has lower individual probabilities of both precipitation and wind events than cluster 4."

[ Question ] At this point, we return to not considering the entire cyclone's life cycle. Each cyclone type has moments of lower pressure in the centre, more significant precipitation, intense winds and higher waves. Likewise, each of these extremes is known to occur at particular times during the cyclone's life. So, I wonder if the compounding probability's magnitude is relevant or just its existence is enough. In other words, does defining the compounding probability from 0 to 1 give us information different from true or false?

[Response] Yes, compounding probability from 0 to 1 gives information that is different from a simple true or false. First, it allows to quantify the horizontal variations in risk around the storm center, among the points where risk is statistically different from the average. Among those points, probability varies by up to a factor 10 (e.g., Fig. 9). Even within an imperfect analysis, as all analyses are, this can be considered valuable information. Second, as we have now shown (in the new Fig. 8, included at the end of this document), peak multivariate risk does occur around the time of minimum central pressure. The reviewers' comment may also pertain to temporally compounding events (successive maxima in winds and precipitations), which are a different type of event altogether and are not discussed here.

[ Lines 255-256 ] "As a result, the rain probabilities shown here are slightly different than the ones shown in Fig.4."

[ Correction ] This information would be relevant if we knew the number of events identified and the number of years that the study covers. As readers, we have no idea whether these numbers are relevant on a climatological scale or represent ten years when the Mediterranean Sea SST was extraordinarily high, for example.

[Response] The composite masking procedure is applied to the same time period as non-masked composites, not a separate time period, and we believe, as stated in the text, that the fact that we constrain the analysis to cyclones peaking over the sea is a more than sufficient explanation for the differences in precipitation (see for examples the clusters that simply have no precipitation associated after the masking). As for the number of years covered by the study, they are now stated earlier in the text (1980-2019 for these composites). Still, to clarify the effect of the masking on the number of storms accounted for, we added the text:

"The masked composites for clusters 2, 4 and 8 include respectively 65%, 52% and 39% percents of the total number of cyclones in the clusters, distributed between 1980 and 2019."

[ Line 266 ] "Note the case of cluster 2,4? ..."

[ Correction ] I think something is missing here.

[Response] Indeed there was a typo. It should have been "cluster 4", and that has been fixed.

[ Lines 268-269 ] "In fact, none of the clusters most important for wind and rain compositing (2, 4 and 1) are associated with heat event field significance."

[ Question ] What correlation with cyclones would the authors like to highlight in analysing the pm10-heat? From the manuscript's beginning, it is unclear how this hazard composition is associated with cyclone passages. In theory, the most significant risks concerning this composition are expected to occur specifically in the absence of cyclones.

[Response] We specifically wish to highlight whether the passage of a cyclone is associated with risk that is significantly elevated with respect to background seasonal values. The results (e.g., Figure 11) show clearly that this is the case, and that dust is particularly elevated while cyclones are present. Hence we are not sure why the reviewer states that these conditions should occur particularly in the absence of cyclones.

[ Figure 4 ] ...

[ Suggestion ] Consider presenting Figures 4, 5, and 6 in a single figure to facilitate comparing these results and following the analysis described in the manuscript.

[Response] While we appreciate the reviewer's point, we prefer keeping these figures separate to enable closer proximity between the figures and the text that describe them, as well as leaving enough space for each column.

[ Lines 279-280 ] "... the density plots capture similar high-density 280 areas as Fig.1, which represented the density of tracks at all times."

[ Correction ] This is mandatory as your data has been filtered for cyclone tracks.

[Response] This is expected but not mandatory since this figure does not show the same results as Fig.1. As stated in the text, Fig.1 shows "the average number of storm centers tracked at a given location per year", while Fig.7 shows "the spatial density and seasonal cycle of storms at the time of minimum central pressure". While that is not likely, one could imagine a maximum of storms tracks transiting through a given location, and a distinct maximum where storms reach their peak intensity. In order to clarify this point a bit, we added "As can be expected" to the beginning of the sentence.

[ Lines 283 ] "... and craters to zero in mid-to-late summer."

[ Correction ] To provide a more accurate understanding, let's highlight that the dataset used to track cyclone tracks has certain limitations during the summer months. These limitations arise due to the need to converge different methods to identify and track cyclones. By acknowledging these limitations, we can work towards finding more effective solutions to track cyclonic activities accurately in all seasons.

[Response] The limitations were acknowledged in response to a relevant previous comment by the reviewer. The idea here is to compare the seasonal cycles of different storm clusters, some of which do peak in the summer. However, we agree that stating that the number of tracks goes to zero is overconfident, given the inaccuracies of the tracking during that season. Hence we replaced that part of the sentence with "... reaches a minimum in mid-to-late summer."

[ Lines 296 ] "(Givon et al. 2023)"

[ Correction ] If the results in Figure 8 were not produced by Givon et al (2023), there is no reason to cite this study.

[Response] Agreed, that has been corrected. In fact, the whole figure was removed as part of responding to another comment.

[ Figure 7 ] ...

[ Correction ] Name the colour scale units and, if possible, enlarge the whole figure.

[Response] Thank you for the comment. The relative density is unitless, and the figure has been enlarged. We have added some clarification in the figure caption: "Relative spatial density of cyclones at the time of minimum central pressure [...]"

And in the figure description: "The relative density is computed by dividing the number of cyclones occurring over a given area by the total number of cyclones in the clusters

considered."

[ Lines 303 ] "... cluster 8 ORM TO ME ALSO ACWB, CL7 LOOKS CYCLONIC may be associated ..."

[ Correction ] Check this sentence, please.

[Response] Thank you for noticing. That must be a comment that made its way unedited in the text. It has been corrected.

[ Figure 8 ] Caption "... potential? sea-level pressure"

[ Correction ] Check this sentence, please.

[Response] Thank you for noticing. Yet another leftover of text edition. It has been corrected.

[ Figure 8 ] ...

[ Correction ] Name the colour scale units, add numbers to the black contours, and, if possible, enlarge the whole figure.

[Response] The entire figure has been removed and replaced, in appendix, by a reproduction of the Figure of Givon (2023) over which this figure was based.

[ Lines 323 ] "... occurs at a given location and in a given cluster."

[ Correction ] This given location does not refer to a random one but to a specific position around the cyclone's centre at the moment of lowest pressure. It is essential to highlight this, as this result does not apply to all situations.

[Response] We agree and have added "at the time of minimum pressure." at the end of the sentence.

[ Lines 338-340 ] "In general, compound risk peaks to the west-north-west of the storm center, with a crown of (lower) risk encircling the storm center. Despite having somewhat smaller and weaker wind and rain footprints than cluster 4, cluster 2 is associated with the highest rain-wind compound risk."

[ Question ] Considering that Mediterranean cyclones present a wide range of trajectories which often change direction during their life cycle, how generic can we consider Fig9 for cyclones at their moment of lowest centre pressure? Remember that the most remarkable precipitation areas are usually downwind of the cyclone's centre and trajectory. For example, can we expect that Fig9 adequately describes cyclones rising at the lee of the Alps and those with cyclogenesis along the Libyan coast, even if both head to the Ionian Sea? In this case, we expect the maximum precipitation surrounding the southern area of the Alps cyclone's centre and the northern area of the Libyan cyclone.

[Response] We agree that the direction of the trajectory may influence the relative position of the storm around the cyclone center. However, the fact that the hazard region is highly concentrated in a specific location relative to the center, by opposition to diffuse around it, indicates that this effect is not very important here. Hence, we chose to retain the same approach to compositing.

[ Lines 350-351 ] "In clusters 2 and 4, the wave event footprints tend to extend to the northeast of the storm center."

[ Correction ] This information is incorrect. Cluster 2 shows a wave's probability peak westward, while Cluster 4 peaks southwest.

[Response] We do not mention the probability peak here, but the significant wave event footprint, so this sentence is not incorrect. We do so because even if events are not peaking to the northeast of storm centers, their presence there is collocated with precipitation in a way that the peak wave event region is not. Hence these events are more important for compounding. Nevertheless, to make this clearer to the readers, the sentence has been modified and now reads "In clusters 2 and 4, the wave event footprints tend to extend to the west, but also to the northeast of the storm center."

[ Figure 9 ] Caption "The black contour identifies the statistically significant area for each hazard."

[ Correction ] Add the threshold used.

[Response] This threshold if not fixed but, as stated in Eq.1, it is necessarily smaller than or equal to 0.05. Hence we have added a reference to Eq.1 in the caption.

[ Figure 10 ] Caption "The black contour identifies the statistically significant area for each hazard."

[ Correction ] Add the threshold used.

[Response] This threshold if not fixed but, as stated in Eq.1, it is necessarily smaller than or equal to 0.05. Hence we have added a reference to Eq.1 in the caption.

[ Figure 11 ] Caption "The black contour identifies the statistically significant area for each hazard."

[ Correction ] Add the threshold used.

[Response] This threshold if not fixed but, as stated in Eq.1, it is necessarily smaller than or equal to 0.05. Hence we have added a reference to Eq.1 in the caption.

[ Line 406 ] "... a heat event is co-occurring."

[ Suggestion ] Given that heat events were defined by a fixed threshold (30°C) instead of a percentile based on local temperatures and the clusters related to the pm10-heat composition occur over North Africa, every analysed event is a heat event, making the probability of occurrence of this hazard composition almost exclusively dependent only on PM10. Maybe you should mention it.

[Response] We agree that the occurrence of this hazard composition is more dependent on a single hazard than the wind-rain or wave-rain events, but we strongly disagree that it is almost exclusively the case. The probability of compound event in cluster 5, for example, is less that a quarter of the probability of PM10 event. Furthermore, both cluster 7 and cluster 5 exhibit a lower compound event probability than cluster 6, despite having more frequent dust events. The situation described by the reviewer only seems to be the case in cluster 6, which was already acknowledged in the text: " This is due to the very high warm spell probability associated with heat lows over the desert in that cluster, which means that most of the time a PM10 event occurs, a warm spell is co-occurring."

[ Line 407 ] "... Simultaneity and Overlap, with two interesting exceptions."

[ Question ] Is there any explanation for this?

[Response] We have added a possible explanation for this results:

"The latter may be due to the fact that cluster 9 has a broad geographical distribution, from the lee of the Alps to the Sahara Givon et al. (2023), which may decrease the incidence of both individual and compound hazards without impacting Simultaneity or Overlap. Within that cluster, cyclones occurring over the desert may only be associated with warm spells and PM10, not rain, while the reverse may be true for one occurring over the sea, but both may be associated with high Simultaneity and Overlap locally."

**Reviewer 2**

We thank the reviewers for their very insightful and valuable comments, and for all of their work in helping to improve this paper. We hope that the improvements to the paper will correspond to the reviewers vision and expectations.

General comments: This analysis provides a unique perspective on compound hazards. Linking the hazards to the large-scale circulation systems provides an opportunity to quantify the links between different upper-level forcing within Mediterranean cyclones and the hazards that are generated. The analysis plan is clear.

Major comment:

As I read the manuscript, I was stuck on the question: what does this analysis provide? I think this statement in the abstract is nice:

Line 6: "few different large-scale configurations lead to each compound event type."

Thus, the work is reducing the complexity of the compound hazards, or at least giving some useable information about probabilities. However, this next line in the abstract in particular made me wonder, what are the authors intending to demonstrate:

Lines (10-13) Next, we find that the probability of compounding associated with a cyclone class does not depend monotonically on the probabilities of the individual contributing hazards, but also on the goodness of their temporal and spatial correspondence.

What do you mean by "goodness of their temporal and spatial correspondence". Is goodness quantified in the manuscript? I did not see that it was. Related to this: it seems to me, the 3 potential uses of the climatology that the authors list on lines 58 – 63 in the introduction should be the elements that are highlighted in the abstract and the conclusion. Perhaps, the sentence on lines 10 -13 is somehow linked to those 3 uses, but I was not able to understand the connection.

The constructive point that I am trying to make here is this: the abstract and the conclusion need to do a better job in demonstrating the utility of this work.

We agree that the abstract was unclear. "goodness of their temporal and spatial correspondence" was indeed undefined and was an inadequate way to refer to the Simultaneity and Overlap metrics introduced in the paper. Most of the abstract has been restructured to try and better summarize the work and demonstrate its utility:

"Cyclones are responsible for much of the weather damage in the Mediterranean region, and while their association with individual weather hazards is well understood, their association with multivariate compound hazards remains to be quantified. Since hazard compounding is associated with enhanced risk, this study aims to establish a cyclone-relative climatology of three different multivariate hazards in Mediterranean cyclones, namely, the co-occurrences of rain and wind, rain and wave, and particulate matter and warm spells. The hazards are composited separately for nine cyclones classes associated with nine large-scale environments, using a recent cyclone classification. This cluster-based compositing of multivariate hazards outlines the role of the large-scale environment in the occurrence of impactful cyclones. The composites are computed relative to cyclones centers and at the time of maximum intensity, when the association with compound hazards is strongest for most of the nine cyclone classes, to illustrate the spatial footprint of the multivariate hazards associated with the cyclones. Finally, datasets of cold fronts, warm conveyor belts and dry intrusions are composited alongside the hazards to provide information on the contribution of smaller-scale features to the occurrence of multivariate hazards. We find that few different large-scale

configurations are associated with each specific compound event type. Compound rain and wind events are mostly associated with frontal cyclones and cyclones induced by anticyclonic Rossby wave breaking. These events are most frequent in the winter half-year. Compound rain and wave events occur also primarily during winter, but are also associated with cyclonic Rossby wave breaking. Particulate matter and heat compound events are associated with heat lows, daughter cyclones and anticyclonic Rossby wave breaking in the warm season and over North-Africa. The probability of compounding associated with a cyclone class does not depend monotonically on the probabilities of the individual contributing hazards, but also on their temporal and spatial correspondence. Finally, we find warm conveyor belts and cold fronts to frequently co-occur with rain and wind, and rain and wave events. The association of compound hazards with warm conveyor belts and cold fronts is similar to previous results from the Atlantic basin but substantially modulated by the local topography and land-sea distribution. Particulate matter and warm spells are not strongly associated with these dynamical features. These results, which systematically associate various large-scale environments and dynamical features to different compound event types, have implications for forecasting and climate risk predictions."

We also added clarifications to the conclusion: "To summarize, this study established multivariate hazard footprints relative to Mediterranean cyclones for various large scale configurations and in relation to cyclone dynamical features, aiming to provide information on cyclone-relative hazard occurrence, and on the contribution of various scales and processes to that occurrence."

Minor comments:

There are some glaring editing issues (listed below), that give the impression that the final phase of work before submission was a bit rushed, which is too bad.

Line 6, and elsewhere: if "large scale" is describing something and comes before that thing, it needs to be hyphenated. You have it correct in some places, but in others, such as line 102, it needs to be hyphenated.

We thank the reviewer for noticing the issue, and have corrected it throughout the text.

Line 29-30: (This is just a terminology question I think). You write the "multivariate hazards" are defined as those that lead to greater impacts than any individual hazard would cause. How are impacts defined in this context? And does this make "multivariate hazards" a subset of compound hazards? Compounds hazards being an multi-hazard event and a multivariate hazard being a compound hazard in which specific details about the impacts are known. If I am interpreting what you've said on Lines 29-30 correctly, then are you studying multivariate hazards? i.e., do you actually have impacts data in the current analysis? it seems to me that you are studying the characteristics of the hazards, not the impacts. So, either, the term multivariate hazards does not have direct implications about the impact of a

compound event, or, you are not studying multivariate hazards, you are studying compound hazards.

We agree that this needs to be clarified. Multivariate hazards are indeed a subclass of compound hazards, and all compound hazards, including multivariate hazards, are known for leading to enhanced risk. Our statement that compound events lead to greater impacts was not correct. These events MAY lead to greater impacts but they should be defined by the increased risk they pose. We attempt to clarify these points by re-framing the paragraph:

"Multivariate compound events, defined as events where the co-occurrence of hazards leads to enhanced risk of impacts by comparison to any individual hazard (Zscheischler et al. 2020), have been shown to occur in the Mediterranean. [...] In this paper we will use the terminology multivariate hazards and multivariate compound events interchangeably to indicate the co-occurrence of hazards that have been shown by prior literature to lead to enhanced risk in the region."

In other words, we are not evaluating impacts in the paper, but only the occurrence of conditions known to be associated with a higher risk and known to potentially lead to higher impacts in the Mediterranean Basin.

Line 77: I could not find anywhere in the methods in which you state the length of the records being used in this analysis. I assume the PM10 is the limiting dataset in terms of going back in time.

Thank you for the comment, that is correct. The period used was indeed determined by the largest overlap of the datasets available, and the shortedt dataset was indeed PM10. Hence, we specify in section 2: "Since all datasets, except PM10, are available for the period 1980-2019, all analyses cover that period, except for PM10-related analyses, which extend from 2004 to 2019."

Line 101: You have written "Figs. X and Y of Givon et al. 2023". Presumably X and Y are placeholders that need to be updated with actual numbers. (Editing issue #1).

Thank you for the comment, the figure numbers were indeed missing and have been substituted in for the X and Y.

Lines 110 – 115: I appreciate that you spent some time giving context the nature of the precipitation data provided by reanalysis – as well as a citation regarding its skill. Please do the same for wave heights.

We agree and have added some context for the limitations of the swh data: "The reanalysis swh also compares favorably to observations, albeit ERA5 may underestimate extreme wave heights near the coasts (Fanti et al., 2023), which are of interest here."

Line 132: I am not sure what to make of this citation. I've never encountered embargoed theses citations before. Perhaps this is fine, but I thought it best to flag it with a question.

Fortunately, the embargo was lifted in February 2024, and the publication is now properly cited.

Line 152: For this map, why are did you choose to show such a large domain, it seems more appropriate to use the same domain as in Fig 1. Or perhaps I have skipped over some detail in which you explain the reasoning.

Since we allow for a large compositing domain (40 x 40) around the storm center, the domain over which hazards thresholds need to be defined must be considerably larger than the domain over which tracks occur (here by 20 degrees).

Line 175: Using a 40x40 degree grid box seems a bit excessive, especially if you are wanting to focus on compound events in the same region. This choice is also made a bit mysterious by the fact that the averaging you do later is over a 20 x 20 region (line 225). Please explain these choices.

Indeed, this choice needs to be explained: we aimed to reproduce the PV structure composites of Givon et al. 2023 for validation, which require this large averaging region, even if the hazards themselves do not. In the end, the figure representing the PV structure was removed, but the rationale of taking a larger-than-necessary box to allow for analyses requiring a larger domain remains. We have added the sentence: "This large box was chosen because the scale of the significant hazard fields is not a priori known for all hazards, and to produce validation composites of the large-scale environment (not shown)."

Line 176: For Mediterranean cyclones, have previous studies demonstrated that the time of minimum SLP corresponds to the time of strongest winds or precipitation? If so, please cite here. If not, I don't think additional work is required for the present study, but there should be some comment in the manuscript about the assumption you are making.

This is an important comment, which echoes the concerns of other reviewers, and we have attempted to address it thoroughly while remaining within the scope of the paper. Once the clusters of interest are identified (2-4-1 for Wind-Rain, 2-4-8 for Wave-Rain and 6-7-5 for PM10-Heat), we present a figure which shows the time evolution, 6 hourly, of the composite multivariate hazard likelihood, centered on the time of minimum central pressure. This new figure (Fig. 8, included at the end of this document) shows that the time of minimum pressure is associated with the maximum compound hazard likelihood in most clusters. Even in the clusters where another time is associated with higher hazards, the difference is small and we think the time of minimum pressure is still representative of the hazard maximum (described in Sect. 3.3). Hence, to make sure that the composites remain easy to interpret, we keep on using the time of minimum central pressure as a unique compositing time step,

but that choice is now better justified.

Line 240 (Fig. 4): Figure 4 alone was not that useful to be until I looked at Figure 3 from Givon et al. (2023). I see now that your Fig. 8 is a subset of Givon et al. Fig 4, and I just wonder if you want to show your Fig. 8 before Fig. 4? Or, I wonder if you want simply include the figure 3 from Givon et al., if this journal allows for that.

This is a point we've had to give some thought to. The paper is fairly lengthy and, to avoid extending it further, we aimed to avoid discussing extensively all clusters and their dynamical context (which is also already done by Givon et al. 2023). Hence, the way we chose to narrow down the analyses and to group clusters (for example for the study of seasonality) was to introduce first the hazard summary figures and only afterwards the geographical, seasonal and dynamical context. The reason for this being that we are interested in the context within which the multivariate hazards occur more so that the context within which all cyclones occur.

However, the reviewer raises a valid point. To respond to it, we have removed Fig. 8 altogether and have reproduced Fig. 3 of Givon et al. (2023) in the appendix of the present paper, so that all clusters are now represented, but it is made clear it is not a contribution from our paper. The PV structures are now discussed in two locations in the paper, including one much earlier. The first one is when the cluster classification is introduced (Sect. 2.1), so that the readers can interpret the results of Fig.4 right away. And the second one is when the seasonal and geographical variations of the clusters of interest are discussed (Sect. 3.2).

Line 303: There seems to be a note between co-authors here that did not get edited out. (Editing issue #2).

Thank you for the comment, this has been corrected.

Line 317: The caption for Fig. 8 has an editorial error, you write:

"the surface environment is represented by contours of potential? sea-level pressure (black contours at 250 Pa intervals)." (Editing issue #3).

Thank you for the comment, this figure was removed and replaced, in appendix, by the original figure of Givon et al. (2023).

Line 367: Fig. 9 could be edited so that the differences between the 3 columns are easier for the viewer to find. First, I suggest decreasing the extent of the x- and y-axes to +/-10 degrees maximum. Second, I would tighten or change the colorbar scale in the first and third column. Right now, there is just too much white space in these sub-panels.

Axis extent has been adjusted to 10 degrees, but we have elected to keep the colorbar scales identical as we believe they allow the reader to compare the maxima and minima of the

hazard probabilities within the statistically significant region. The white space has been considerably reduced by adjusting the axes, and features are now easier to see.

Figs. 10 and 11 could use some of the same changes. In Fig 10, perhaps the colorbar range is ok, but the x- and y-axes are extended too far. In Fig 11, the colorbar range for heat only is too narrow isn't it?

Figures 10 and 11 have been adjusted according to the recommendations of the reviewer.

Line 370: (Section 4 Discussion): I was expecting some discussion of the relative occurrence of hazards in the absences of cyclones, but I don't see it, or I missed it. I suggest that the work would benefit from some further analysis, or at the least some discussion regarding the hazards that are not associated with cyclones.

We agree with this comment and have added a section to the discussion, entitled "Comparison with conditions in the absence of storms." The additional text follows:

"In this paper, we have focused on the frequency of compound hazard within storms, but not on the general occurrence of compound hazards in the region. This endeavor is outside the scope of this paper and is addressed for Rain-Wind events in Portal et al. (2024). Taking an Eulerian perspective over the Mediterranean Basin, they show that locally, Rain-Wind event frequency is enhanced by a factor of at least 5 (and sometimes much more) by the presence of a cyclone. In the present study, examining storm-relative composites (Figs.9, 10 and 11) provides some insight on the increase of hazard likelihood brought about by the presence of the storms, in comparison to that at the same locations and for seasonally similar times. The relative difference between the maximum and the minimum event probabilities within the statistically significant region gives a sense of the influence of the storm with respect to background value. For example, there is roughly a factor 10 between the probability minima and maxima within the statistically significant regions in Fig. 9, and those minima still represent probabilities that are significantly higher than the background. There is somewhat less difference between the maxima and minima for the rain-wave events in Fig. 10, due to the wave background values being high. The relative difference between maxima and minima is yet again smaller for PM10 events, and hence PM10-heat events (Fig. 11), because the cyclones in these clusters occur over the Sahara (see Fig. 7), a part of the world where dust emissions are particularly high.

**Reviewer 3**

We thank the reviewers for their very insightful and valuable comments, and for all of their work in helping to improve this paper. We hope that the improvements to the paper will correspond to the reviewers vision and expectations.

The manuscript by Rousseau-Rizzi et al. discusses a system-relative climatology of compound events associated with Mediterranean cyclones focusing on wind-rain, rain-waves, and particulate matter-heat events. The study shows that compound events often happen during specific clusters, which is also helpful knowledge for forecasting future events. Overall, the work shows great value and is well written.

However, I have two concerns discussed below that I think need attention before publishing.

Main concerns 1. Only using one time step per cyclone: I understand that only using one time step simplifies this work tremendously. However, some of these hazards and dynamical features discussed occur during different times of a cyclone's life cycle (e.g., Hewson and Neu (2015) and more recently Eisenstein et al. (2023), although these works have been done over the North Atlantic and Central Europe). I see problems only using the time of minimum pressure as it will exclude a lot of information and hazards neglecting the development of a storm. Maybe some compound events happen sooner or later in the life cycle? Even though I understand how much work it would be to include more time steps, especially considering possible double-counting of cyclones as you mentioned, I believe this needs to be included to make this work even more meaningful. At least a detailed discussion about what would change, looking at other time steps with a few examples, must be added. Also, how important are hazards that are detected immediately after one another?

This is an important comment, which echoes the concerns of other reviewers, and we have attempted to address it thoroughly while remaining within the scope of the paper. Once the clusters of interest are identified (2-4-1 for Wind-Rain, 2-4-8 for Wave-Rain and 6-7-5 for PM10-Heat), we present a figure which shows the time evolution, 6 hourly, of the composite multivariate hazard likelihood, centered on the time of minimum central pressure. This new figure (Fig. 8, included at the end of this document) shows that the time of minimum pressure is associated with the maximum compound hazard likelihood in most clusters. Even in the clusters where another time is associated with higher hazards, the difference is small and we think the time of minimum pressure is still representative of the hazard maximum (described in Sect. 3.3). Hence, to make sure that the composites remain easy to interpret, we keep on using the time of minimum central pressure as a unique compositing time step, but that choice is now better justified.

To address the comment about hazards occurring sequentially, we have added the following text to the discussion: "Multivariate compound events, which are the focus of this paper, require that the different hazards involved should occur at the same time and location, and these requirements determine the definition of Simultaneity and Overlap. The probability of other types of compound events, such as temporally compounding events where the risk increases due to hazards occurring in a succession (Zscheischler et al. 2020), cannot be understood using these metrics. In addition, the hazards studied here have been selected specifically because they lead to multivariate events and these hazards are not all prone

to leading to increased impacts when occurring sequentially. For example the enhanced damage to infrastructure due to the rain and wind co-occurring, also called Wind-Driven Rain, depends on the rain falling at an angle because of strong winds occurring at the very same time (e.g., Blocken and Carmeliet, 2004). Wind occurring after rain, or vice-versa, would not lead to the same risk enhancement. Hence, the future studies of different types of compound hazards will require the development of new metrics."

2. Dynamical features: I am missing a clear description of the dynamical features, their hazards (wind, precipitation), etc. There are numerous studies of the features in extratropical cyclones (e.g., regarding wind: Hewson and Neu, 2015; Clark and Gray, 2018; Eisenstein et al., 2022,2023; regarding precipitation: Catto and Pfahl, 2013; Catto, 2016, . . . ). Do they differ in the Mediterranean/how so? For example, regarding the WCB: Clarify whether you are only interested in winds(warm conveyor belt jet; see literature above) or also in the forming cloud head which is responsible for most of the precipitation (Catto, 2016), which I assume as you consider it up to a height of 400hPa. Maybe consider explaining the difference of the warm conveyor belt jet, responsible for the high surface winds, and the WCB forming the cloud head and precipitation. I believe a more in-depth explanation and discussion is needed, especially for readers less familiar with these dynamical features.

We agree that additional context was needed. To start, we added specific information on reported relations between the dynamical features and the hazards considered, either in the Mediterranean or in other regions of the world:

"CFs are associated with 75% of the precipitation extremes in the midlatitudes (Catto and Pfahl, 2013) as well as with extreme winds (Dowdy and Catto, 2017; Catto and Dowdy, 2021), WCBs have been shown to be associated with extreme precipitation in the Mediterranean (Flaounas et al., 2018) and wind and wave near the British Isles (Gentile and Gray, 2023), and DIs are associated with enhanced surface wind gusts globally (Raveh-Rubin, 2017). Cyclone dynamical features and their relation to hazards in the Mediterranean may differ from those in other regions of the world because of the strong modulation of the cyclones by topography and by the land-sea distribution, and because of the distinct structures of the cyclones in the region (Flaounas et al., 2022). For example, case studies have shown lower cloudiness associated with WCBs in the Mediterranean, in cases where dry Saharan air was entrained in the cyclone (Ziv et al., 2010). Hence, quantifying the relation between Mediterranean cyclone dynamical features and hazards provides valuable information on the smaller-scale processes involved in the occurrence of hazards."

We then added more information on the feature detection methods employed in the datasets we use in the study:

"The DI identification is based on a systematic airmass trajectory computation, where DI

trajectories are defined by their descending of at least 400 hPa within 48 h. A DI is then considered present at a given grid point and time if any DI trajectory is present there at a pressure level of 700hPa or higher (i.e., in the lower troposphere). Thus, we consider so-called "DI outflows" as referred to in Catto and Raveh-Rubin (2019) and adapted here to ERA5. Similarly, the WCB identification is based on trajectories that ascend 600 hPa or more within 48 h. The density of WCB trajectories is then computed for three different pressure intervals (Heitmann et al., 2023): an inflow interval (between the surface and 800 hPa), an ascent interval (between 800 and 400 hPa), and an outflow interval (pressures lower than 400 hPa). While the methods used to identify DIs and WCBs are similar, the front identification method (intoduced by Berry et al., 2011) is very distinct. The method first identifies points where the gradient of the thermal front parameter (Renard and Clarke, 1965) is zero, and then connects those points to form 1D features. In the method of Catto and Pfahl (2013), cold front are then identified based on their direction."

We also added some discussion of the different regions of the WCB in relations to the hazards, in section 2.4.1:

"Here, WCB trajectories are assumed to be important for surface hazards only when they occur in the "inflow" or "ascent" regions of the conveyor belt (between the surface and a pressure of 400 hPa). The inflow region is expected to be associated with enhanced surface winds, and the ascent region, with enhanced precipitation. In this study, no distinction is made between the two regions considered."

General comment:

• You include datasets based on ERA-Interim and ERA5. Please clarify what inconsistencies -if any - might occur in doing so.

This was the first author's mistake. It has been clarified by the co-authors that all dynamical features datasets had been re-computed from ERA5 data. All datasets used are based on ERA5. The text has been modified to state:

"These include a dataset of objectively identified front lines (Catto and Pfahl, 2013), a dataset of DI trajectory density (Raveh-Rubin, 2017) and a dataset of WCB trajectory density (Heitmann et al., 2023), computed using ERA5. The front and DI datasets were initially introduced based on ERA-Interim data; we are using here an updated version, based on ERA5."

Minor comments: Abstract • l. 5f Please be clearer about what kind of classification and the "few different large scale configurations".

We agree with the reviewer that this needs to be clarified: We have reformulated the abstract to state " The hazards are composited separately for nine cyclones classes associated with

nine large-scale environments, using a recent cyclone PV-based classification. This cluster-based compositing of multivariate hazards outlines the role of the large-scale environment in the occurrence of impactful cyclones."

Introduction • l. 24 and later: You introduce pm10 here for particulate matter. Later (l. 117) you introduce it for particulate matter specifically of size 10µm. Please clarify this already in the beginning.

Thank you for noticing. We have clarified this at the first introduction of the pm10 abbreviation.

• You consider both winds and waves as hazards, so I would suggest including a quick explanation how both influence each other, see e.g., Gentile et al. (2021) and Gentile and Gray (2023; also including dynamical features)

We agree and have added some additional context as suggested: "We note that individual weather hazards may influence each other. For example, growing waves may lead to an increase in surface roughness and a decrease in near-surface wind speeds (Gentile et al. 2021).". Gentile and Gray (2023) has also been cited in support of the association between waves and dynamical features.

• l. 44f: Please refer to these studies directly instead of a not-published thesis.

The thesis is now published (since February 2024), and represents, to our knowledge, the best example of such an endeavour. We have thus chosen to keep this citation.

• l. 79f: Please introduce the abbreviations of all features in the same paragraph (l. 39) and then use these abbreviations throughout the manuscript.

Thank you for the comment, it has been implemented.

Data and Methods • l. 101 Which figures to you refer to by Fig. "X and Y"? I had a hard time following the different clusters. Please include at least a small description with characteristics and a similar figure to Givon et al. (2023) of the clusters in the appendix or supplementary material. Maybe also show and discuss Fig. 8 earlier. This would make it easier for the reader to follow your discussion.

The appropriate figure numbers have been substituted in the placeholder X and Y, which were left by mistake. For the rest of the comment, to address similar concerns by another reviewer, we have removed Fig. 8 altogether and have reproduced Fig. 3 of Givon et al. (2023) in the appendix of the present paper, so that all clusters are now represented, but it is made clear it is not a contribution from the present paper. The PV structures are now discussed in two locations in the paper, including one much earlier on. The first one is when the cluster classification is introduced (Sect. 2.1), so that the readers can interpret the results of Fig.4 right away. And the second one is when the seasonal and geographical

variations of the clusters of interest are discussed (Sect. 3.2).

• l. 114 This sentence is confusing to me. Do you mean you take the average height of the top 33 percent wave heights? Please clarify and rephrase this sentence.

This description was adopted by ecmwf to provide insight on the parameter, but since it can be confusing, we are adopting the approach of describing the parameter more simply along with giving the full definition. All the ecmwf descriptions we are basing ourselves on are available at the following link: https://codes.ecmwf.int/grib/param-db/140229. The new description is:

"The wave variable used here is the "significant wave height" ("swh"), which represents the height between the wave crests and troughs and is defined as the square root of the integral in space and frequency of the surface wave spectrum, times four."

• l.128 What do you mean by distinct? A few more details about the detection methods should be added for understanding.

[Response] We agree and we added supplemental information. Please see the response to Main Comment no2.

• l. 132 Is there more work on this than a not yet published thesis that can already be cited here?

Thank you for the comment. The thesis is now published and cited elsewhere in the manuscript. For this specific citation, we have taken an earlier published paper by the same author (Owen et al. 2021).

• Fig. 2 Consider swapping the subfigures to be consistent with the order of how they have been introduced. Also, add (a) and (b).

The subfigures have been swapped, but we think that adding (a) and (b) is redundant with the careful labelling of the panels.

• l. 161: "30°C"

The typo has been corrected.

• l. 172f See main comment 2: Which hazards associated with fronts are you interested in – wind, precipitation, or both? Describe the hazards associated with the features.

[Response] We are interested in both and have added this supplemental information. Please see the response to Main Comment no2.

• Is Cyclone ID = Storm ID?

The notation was inconsistent and has been corrected.

• l. 206f I would suggest adding a half sentence explaining Monte-Carlo samples and/or include a reference.

More details have been added to the sentence, which now reads: "Those times are selected for random years but within 15 days of the days of year where the storm events occur, to yield the same seasonal distribution as that of the storm sample (e.g. Welker et al., 2014)."

Results • l. 227 "have a points"

Thank you for the comment, this has been corrected.

• l. 240 and throughout the chapter: Use Fig. X and Sect. X.

Thank you for the comment, this has been corrected throughout the paper.

• l. 266 What do you mean with "cluster 2,4?" ?

Thank you for noticing this typo, it has been corrected.

• l. 297 Again, consider putting figures like this that you do not deem important enough for the main manuscript in the Appendix/Supplementary material.

Thank you for the comment, this has now been done (see response to earlier comment on Fig. 8).

• l. 303 I suppose these are some remnants of an earlier version/comments?

Thank you for noticing, this has been corrected.

• l. 328 This seems to be similar to Central Europe then (Hewson and Neu, 2015; Eisenstein et al.,2023). Did you expect differences?

The similarity is dependent on the cluster (1 and 4 are similar, and 2 isn't). We have added some comparisons: "In clusters 1 and 4, the location of the wind maximum is more directly to the south of the storm center than reported for Atlantic cyclones (Field and Wood, 2007, Owen, 2022), which resembles most the wind signature of cluster 2. The wind signature of subsets of Atlantic cyclones occurring over central Europe may be more similar to that of clusters 1 and 4 (Eisenstein et al. 2023)."

• l. 332 "has and intense"

Thank you for noticing, this has been corrected.

• l. 335 and 228 Is this the cold conveyor belt (jet)? Or the precipitation associated with the cloud head/WCB? Please comment on this.

For brevity, we could not consider all possible dynamical feature definitions and elected not to discuss the cold conveyor belt in the paper, but the strong relation between dynamical features and compound events does suggest that the distribution of events around the cyclone may be determined by the presence of dynamical features. Hence, in Section 4.3, we have added:

"The good correspondence between these compound hazards and WCB or CF features suggests that the distinctive distribution of the hazards around the cyclone center (see Figs. 9 and 10) may be modulated by the dynamical features themselves."

• Fig. 9,10,11 Consider using "row" instead of "line". Further, I suggest changing the caption to "Probabilities (shading) [...]."

Thank you for the suggestions. They have been implemented and indeed make the caption more readable.

Discussion • How would events be taken into account that are detected shortly after one another (if you would look at more time steps, main comment 1)?

See response to Main Comment no1

• l. 399 "shows that˙ overlap [. . . ]"

Thank you for noticing, this has been corrected.

• l. 452ff the reader would benefit from a clearer description of the WCB earlier here (see main comment 2)

[Response] We agree and we added supplemental information. Please see the response to Main Comment no2.

• Same paragraph: What about turbulence of the features, affected area etc.

We are unsure what the reviewer means by "turbulence of the features".

• l. 463 How come DIs tend to occur far from the cyclone centre comparing it to the figures of Browning (1997)?

There may be multiple reasons for this, including the fact that the study of Browning presents case studies rather than a climatology and thus cannot be directly compared. In addition, there are substantial differences between Atlantic and Mediterranean cyclones, which are not yet systematically documented. This may be one of these. To offer this hypothesis in the article, we added the sentence:

"The distance between most DI occurrences and the storm center differs from the conceptual

picture presented by Browning (1997), which is based on Atlantic storms and may reflect a weaker association between these dynamical features and Mediterranean cyclones."

Conclusion • l. 506 "where"?

Thank you for noticing, this has been corrected.

**1 Figures**

[Figure]

Figure R1: Time evolution of composites over a 1000 km radius disk around storm centers for Rain-Wind (top row), Rain-Wave (middle row) and PM10-Heat (bottom row) clusters of interest: Average percentage of the area that is occupied by compound hazards (purple dots and line) and fraction of tracked cyclones with compound hazards (gray bars). The fraction of cyclones with compound hazards is represented by the fraction of the total bar height (gray + white) that is occupied by the gray bar.